# Inducible and reversible phenotypes in a novel mouse model of Friedreich's Ataxia

Vijayendran Chandran[1†‡*], Kun Gao[1], Vivek Swarup[1], Revital Versano[1], Hongmei Dong[1], Maria C Jordan[2], Daniel H Geschwind[1,3†*]

[1]Program in Neurogenetics, Department of Neurology, David Geffen School of Medicine, University of California, Los Angeles, Los Angeles, United States; [2]Department of Physiology, David Geffen School of Medicine, University of California, Los Angeles, Los Angeles, United States; [3]Department of Human Genetics, David Geffen School of Medicine, University of California, Los Angeles, Los Angeles, United States

*For correspondence:
vijayendran@ufl.edu (VC);
dhg@mednet.ucla.edu (DHG)

†These authors contributed equally to this work

Present address: ‡Department of Pediatrics, School of Medicine, University of Florida, Gainesville, United States

Competing interests: The authors declare that no competing interests exist.

**Abstract** Friedreich's ataxia (FRDA), the most common inherited ataxia, is caused by recessive mutations that reduce the levels of frataxin (FXN), a mitochondrial iron binding protein. We developed an inducible mouse model of *Fxn* deficiency that enabled us to control the onset and progression of disease phenotypes by the modulation of *Fxn* levels. Systemic knockdown of *Fxn* in adult mice led to multiple phenotypes paralleling those observed in human patients across multiple organ systems. By reversing knockdown after clinical features appear, we were able to determine to what extent observed phenotypes represent reversible cellular dysfunction. Remarkably, upon restoration of near wild-type FXN levels, we observed significant recovery of function, associated pathology and transcriptomic dysregulation even after substantial motor dysfunction and pathology were observed. This model will be of broad utility in therapeutic development and in refining our understanding of the relative contribution of reversible cellular dysfunction at different stages in disease.

DOI: https://doi.org/10.7554/eLife.30054.001

## Introduction

A guanine-adenine-adenine (GAA) trinucleotide repeat expansion within the first intron of the frataxin (*Fxn*) gene in the chromosome 9 is the major cause of Friedreich's ataxia (FRDA), the most commonly inherited ataxia (*Chamberlain et al., 1988*; *Campuzano et al., 1996*). Due to recessive inheritance of this GAA repeat expansion, patients have a marked deficiency of *Fxn* mRNA and protein levels caused by reduced *Fxn* gene transcription (*Campuzano et al., 1996*; *Campuzano et al., 1997*). Indeed, the GAA expansion size has been shown to correlate with residual *Fxn* levels, earlier onset, and increased severity of disease (*Filla et al., 1996*; *Montermini et al., 1997*). The prevalence of heterozygous carriers of the GAA expansion is between 1:60 and 1:110 in European populations, and the heterozygous carriers of a *Fxn* mutation are not clinically affected (*Andermann et al., 1976*; *Harding, 1981*).

FRDA is an early-onset neurodegenerative disease that progressively impairs motor function, leading to ataxic gait, cardiac abnormalities, and other medical co-morbidities, ultimately resulting in early mortality (median age of death, 35 years) (*Koeppen and Mazurkiewicz, 2013*; *Koeppen, 2011*). However, identification of the causal gene led to identification of a significant number of patients with late onset, tend to have slower progression with less severe phenotype and are associated with smaller GAA expansions (*Bhidayasiri et al., 2005*). This shorter expansion enables residual *Fxn* expression (*Li et al., 2015*), thus modifying the classical FRDA phenotype, consistent with other data indicating that *Fxn* deficiency is directly related to the FRDA phenotype

(*Seibler et al., 2005*). Extra-neurologic symptoms including metabolic dysfunction and insulin intolerance are observed in the majority and frank type I diabetes is observed in approximately 15% of patients, the severity of which is related to increasing repeat length (*Martinez et al., 2017*; *Galea et al., 2016*; *Coppola et al., 2009*). The mechanisms by which *Fxn* reduction leads to clinical symptoms and signs remain to be elucidated, but molecular and cellular dysfunction mediated by a critical reduction in *Fxn* levels plays a central role (*Pandolfo, 2002*).

FXN is a nuclear-encoded mitochondrial protein involved in the biogenesis of iron-sulphur clusters, which are important for the function of the mitochondrial respiratory chain activity (*Abrahão et al., 2015*; *Tanaka et al., 1996*). Studies in mouse have shown that *Fxn* plays an important role during embryonic development, as homozygous frataxin knockout mice display embryonic lethality (*Cossée et al., 2000*), consistent with FXN's evolutionary conservation from yeast to human (*Campuzano et al., 1996*; *Campuzano et al., 1997*). Over the past several years, multiple mouse models of frataxin deficiency, including a knock-in knockout model (*Miranda et al., 2002*), repeat expansion knock-in model (*Miranda et al., 2002*), transgenic mice containing the entire *Fxn* gene within a human yeast artificial chromosome, YG8R and YG22R (*Al-Mahdawi et al., 2004*; *Al-Mahdawi et al., 2006*), as well as a conditional *Fxn* knockout mouse, including the cardiac-specific (*Puccio et al., 2001*) and a neuron specific model (*Puccio et al., 2001*) have been generated. These existing transgenic and heterozygous knockout FRDA animal models are either mildly symptomatic or restricted in their ability to recapitulate the spatial and temporal aspects of systemic FRDA pathology when they are engineered as tissue-specific conditional knockouts (*Miranda et al., 2002*; *Al-Mahdawi et al., 2004*; *Al-Mahdawi et al., 2006*; *Puccio et al., 2001*; *Simon et al., 2004*; *Perdomini et al., 2013*). Despite advances made towards elucidating FRDA pathogenesis, many questions remain due to the need for mouse models better recapitulating key disease features to understand frataxin protein function, disease pathogenesis, and to test therapeutic agents.

In this regard, one crucial question facing therapeutic development and clinical trials in FRDA is the reversibility of symptoms. The natural history of the disorder has been well described (*Metz et al., 2013*; *Nakashima et al., 2014*), it is not known how clinical features such as significant motor disability relate to reversible processes (e.g. underlying neuronal dysfunction) or reflect irreversible cell death. It is often assumed that clinically significant ataxia and motor dysfunction reflects neurodegeneration, although this may not be the case. This issue, while critically important for therapeutic development, is difficult to address in patients, but we reasoned that we could begin to address this question in an appropriate mouse model.

Here we report an inducible mouse model for FRDA, the FRDAkd mouse, that permits reversible, and yet substantial frataxin knockdown, allowing detailed studies of the temporal progression, or recovery following restoration of frataxin expression - the latter permitting exploration of disease reversal given optimal treatment (normalization of *Fxn* levels). We observe that *Fxn* knockdown leads to behavioral, physiological, pathological, and molecular deficits in FRDAkd mice paralleling those observed in patients, including severe ataxia, cardiac conduction defects and increased left ventricular wall thickness, iron deposition, mitochondrial abnormalities, low aconitase activity, and degeneration of dorsal root ganglia and retina, as well as early mortality. We identify a signature of molecular pathway dysfunction via genome-wide transcriptome analyses, and show reversal of this molecular phenotype and as well as behavioral and pathological measures, even in the setting of significant disability due to motor dysfunction in FRDAkd animals.

## Results

### RNA interference based in vivo frataxin knockdown and rescue

To investigate the neurological and cardiac effects linked to reduced FXN levels and to create a model for testing new therapies in vivo, we sought to generate mice that develop titratable clinical and pathological features of FRDA. We employed recombinase-mediated cassette exchange for genomic integration of a single copy shRNA transgene (doxycycline-inducible) that can mediate frataxin silencing temporally under the control of the H1 promoter via its insertion in a defined genomic locus that is widely expressed (rosa26 ) (*Seibler et al., 2007*). This allowed us to circumvent the lethal effect of organism-wide knockout, while permitting significant frataxin reduction in all tissues.

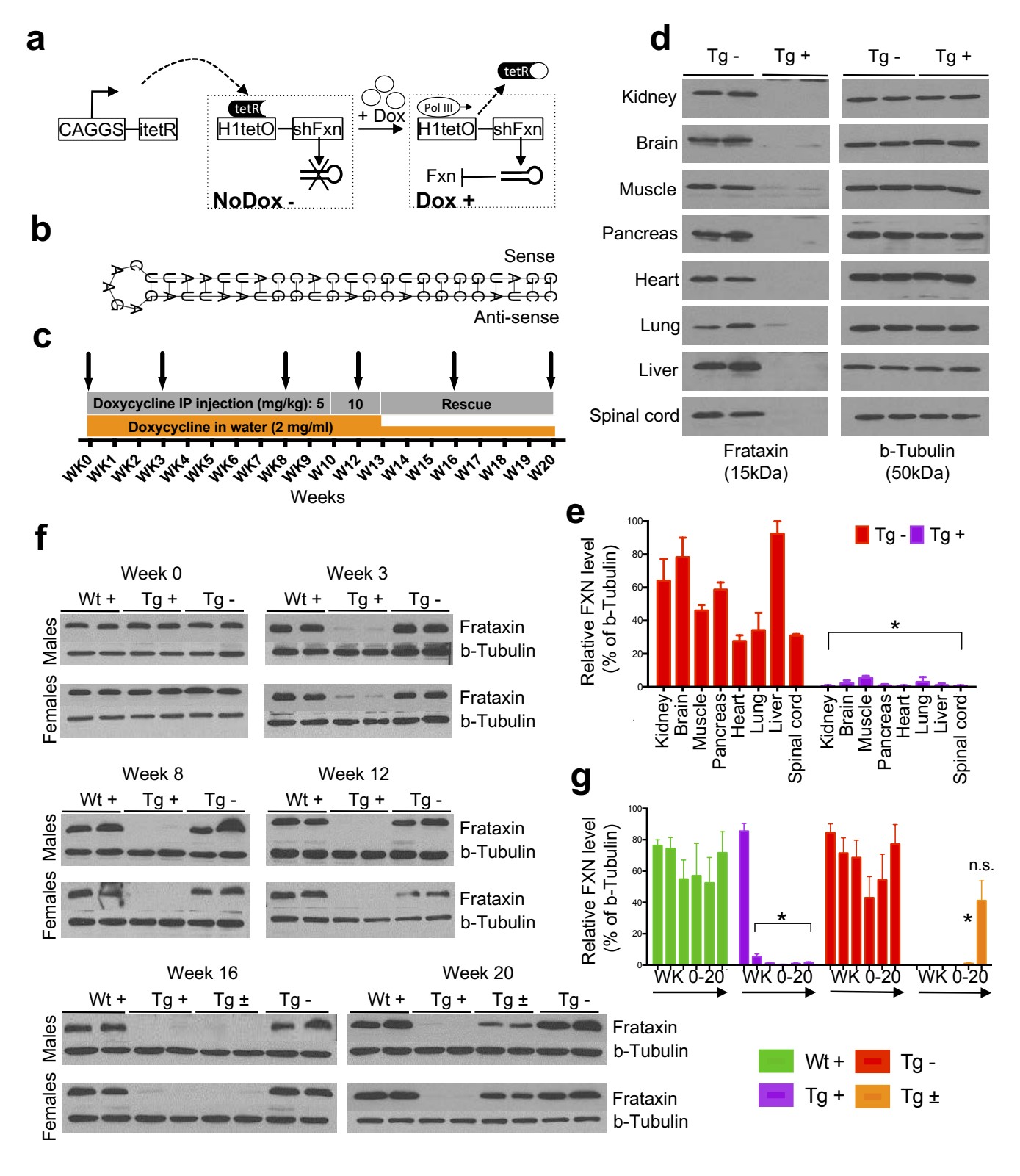

**Figure 1.** Efficient temporal in vivo frataxin knockdown and rescue. (**a**) Schematic representation of the inducible expression vector system delivering shRNA against frataxin. The vector contains an shRNA sequence against the frataxin (*Fxn*) gene regulated by the H1 promoter with tet-operator sequences (tetO) and tet repressor (tetR) under the control of the CAGGS promoter. Transcription of the *Fxn* shRNA is blocked in cells expressing tetR. Upon induction by doxycycline (dox), tetR is removed from the tetO sequences inserted into the promoter, allowing transcription of shRNA against *Fxn*.

*Figure 1 continued on next page*

*Figure 1 continued*

shRNA expression leads to RNAi-mediated knockdown of the *Fxn* gene. (**b**) Predicted minimum free energy secondary structure of expressed shRNA targeting the *Fxn* is shown with the sequence (sense strand) and its complement sequence (antisense strand) in the duplex form along with hairpin loop. (**c**) Timeline for doxycycline treatment of mice with a double IP injection (5 and 10 mg/kg) of dox per week and in drinking water (2 mg/ml). IP injections and dox in water were withdrawn for rescue animals. Arrow signs indicated different time intervals considered for downstream analyses. (**d**) Transgenic mice without (Tg -) or with (Tg +) doxycycline treatment (see c) for 20 weeks were analyzed by Western blot for protein levels of FXN in various organs. (**e**) For quantification, FXN values were normalized to the level of b-tubulin in each lane. (**f**) Time series FXN knockdown and rescue in liver. Wild-type mice with (Wt +), transgenic mice with (Tg +) and without (Tg -) dox, and transgenic mice with (Tg ±) dox removal (rescue) samples were analyzed for weeks 0,3,8,12,16, and 20. Rescue animals (Tg ±) were given dox for 12 weeks and doxycycline was withdrawn for additional 4 and 8 weeks. (**g**) For quantification, FXN values were normalized to the level of b-tubulin in each lane. N = 4 animals per group, *p<0.01; two-way ANOVA test; n.s., not significant. Error bars represent mean ±SEM for panels e and g.
DOI: https://doi.org/10.7554/eLife.30054.002

The following source data and figure supplements are available for figure 1:

**Source data 1.** This spreadsheet contains the Western blot quantification data after frataxin knockdown in various organs and time points in FRDAkd and control animals which was used to generate the bar plots shown in *Figure 1e and g*.
DOI: https://doi.org/10.7554/eLife.30054.005

**Figure supplement 1.** In vitro knockdown validation of *Fxn*-specific shRNAs.
DOI: https://doi.org/10.7554/eLife.30054.003

**Figure supplement 2.** Validation of shRNA off-target effects.
DOI: https://doi.org/10.7554/eLife.30054.004

Depending on the dose of the inductor doxycycline (dox), temporal *Fxn* knockdown was achieved to control the onset and progression of the disease (*Figure 1a*).

Six different shRNA sequences were screened in vitro to obtain a highly efficient shRNA targeting the mature coding sequence of frataxin (*Figure 1b* and *Figure 1—figure supplement 1*). To examine off-target effects, we utilized the shRNA sequence (GGATGGCGTGCTCACCATTAA) to identify potential putative off-target effects in the mouse genome using BLASTN, finding that the second closest match after *Fxn* had only 16 out of 21 bases matching. We observe that *Fxn* is the earliest gene product reduced at the transcriptomic level, and does not alter the expression levels of other potential targets (9 genes with 13–16 nucleotide matches; *Figure 1—figure supplement 2*), consistent with the shRNA's specificity for *Fxn*.

Transgenic animals (FRDAkd) containing a single copy of this efficient shRNA transgene (*Figure 1b*) were generated and characterized. First, to test *Fxn* knockdown efficiency, we explored the response to dox at varying escalating doses in drinking water. At the higher doses, we observed mortality as early as two weeks and a 100% mortality rate by five to six weeks, not permitting extended time series analyses ( Materials and methods). We found that the combination of 2 mg/ml in drinking water coupled with 5 or 10 mg/kg intraperitoneal injection of dox twice per week led to efficient *Fxn* knockdown within two weeks post treatment initiation, while avoiding a high early mortality rate (Materials and methods). Hence, for all subsequent experiments we utilized this regimen to model the chronicity of this disorder in patients by balancing the gradual appearance of clinical signs and decline in function, while limiting early demise (*Figure 1c*).

To determine the effect of *Fxn* deficiency in adult mice after a period of normal development (similar to a later onset phenotype in humans), which would allow establishment of stable baselines, and to obtain relatively homogeneous data from behavioral tests (*Crawley, 2007*), we initiated dox at three months. Following 20 weeks with (Tg +) or without (Tg -) doxycycline administration (*Figure 1c*; Materials and methods), we observed highly efficient silencing of *Fxn*, reaching greater than 90% knockdown across multiple CNS and non-CNS tissues (p<0.05, two-way ANOVA; *Figure 1d,e*). Using this regimen, time series western blot analyses of 80 independent animals (wild-type with dox (Wt +) N = 24; transgenic with dox (Tg +) N = 24; transgenic without dox (Tg -) N = 24; transgenic with dox removal (Rescue Tg ±) N = 8, at weeks 0, 3, 8, 12, 16, and 20) confirmed efficient silencing as early as three weeks, and efficient rescue, as evidenced by normal frataxin levels, post eight weeks dox removal (p<0.05, two-way ANOVA; *Figure 1f,g*). Together, the results indicate that FRDAkd mice treated with dox are effectively FXN depleted in a temporal fashion and that *Fxn* expression can be reversed efficiently by dox removal, making it suitable for studying pathological and clinical phenotypes associated with FRDA.

## Frataxin knockdown mice exhibit neurological deficits

The major neurologic symptom in FRDA is ataxia, which, in conjunction with other neurological deficits including axonal neuropathy and dorsal root ganglion loss, contributes to the gait disorder and neurological disability (*Koeppen and Mazurkiewicz, 2013*; *Koeppen, 2011*). So, we first determined whether *Fxn* knockdown impacted the behavior of Tg and Wt mice with (+) or without (-) dox, or with dox followed by its removal (±; the 'rescue' condition), leading to a total of five groups subjected to a battery of motor behavioral tasks (each group N = 15–30; total mice = 108; Materials and methods) (*Figure 2a–g*). Wt - and Wt + control groups were included to access baseline and to have a control for any potential dox effect on behavior; Tg - and Tg + transgenic groups were compared to examine the effect of genotype and *Fxn* knockdown on behavior; The Tg ± group was included to study the effect of FXN restoration after knockdown (rescue). We observed significant weight loss and reduced survival ratio (<90%) at 25 weeks with dox treatment in *Fxn* knockdown animals (Tg +) compared to other control groups ($p < 0.05$, two-way ANOVA; *Figure 2a,b*). Tg + mice exhibited a shorter distance travelled at both 12 and 24 weeks in comparison to control animals, consistent with decreased locomotor activity ($p < 0.05$, two-way ANOVA; *Figure 2c*). Next, we assessed gait ataxia (*Koeppen, 2011*) using paw print analysis (*Dellon and Dellon, 1991*) (Materials and methods). The *Fxn* knockdown mice (Tg +) displayed reduced hind and front limb stride length when compared with Tg-, as well as the Wt control + at 12 and 24 weeks, suggesting ataxic gait ($p < 0.05$, two-way ANOVA; *Figure 2d,e* and *Figure 2—figure supplement 1*). Grip strength testing also showed that Tg + animals displayed defects in their forelimb muscular strength at 12 and 24 weeks when compared with other groups ($p < 0.05$, two-way ANOVA; *Figure 2f*). Finally, motor coordination and balance were evaluated using the Rotarod test. Whereas no significant difference in time spent on the rod before falling off was seen between Wt + or Wt - or Tg - and Tg ± mice after 12 weeks post dox removal (rescue), chronically treated mice (Tg +mice) from 12 weeks onward fell significantly faster, indicative of motor impairments ($p < 0.05$, two-way ANOVA; *Figure 2g*). These observations suggest that the knockdown of *Fxn* in mice causes motor deficits indicative of ataxia similar to FRDA patients (*Koeppen, 2011*), and demonstrates the necessity of normal levels of *Fxn* expression in adults for proper neurological function.

## Frataxin knockdown leads to cardiomyopathy

Cardiac dysfunction is the most common cause of mortality in FRDA (*Tsou et al., 2011*; *Smyth, 2005*). To examine impaired cardiac function in FRDAkd knockdown animals, we employed electrocardiogram (ECG) and echocardiogram analyses to measure electrical activity and monitor cardiac dimensions. The ECG of Tg + mice displayed a significant increase in QT interval duration when compared to the other control/comparison groups at both 12 and 24 weeks post dox treatment, suggesting abnormal heart rate and rhythm (arrhythmia) (*Surawicz and Knoebel, 1984*) (Materials and methods; $p < 0.05$, two-way ANOVA; *Figure 3a–b,e*). However, rescue animals (Tg ±) 12 weeks after dox removal showed a normal ECG, demonstrating that by restoring FXN levels, the prolonged QT interval can recover ($p < 0.05$, two-way ANOVA; *Figure 3b,e*). We also observed that the Tg +animals at week 24 displayed absence of P-waves, suggesting an atrial conduction abnormality (*German et al., 2016*) (*Figure 3c*), not observed in the rescued animals. Similar cardiac abnormalities have been variably observed in FRDA patients (*Dutka et al., 1999*).

Progressive hypertrophic cardiomyopathy (thickening of ventricular walls) related to the severity of frataxin deficiency (*Dutka et al., 1999*; *Isnard et al., 1997*; *Dürr et al., 1996*) is frequently observed in FRDA patients (*Tsou et al., 2011*). To confirm structural cardiac abnormalities, echocardiogram analyses were performed, focusing on left ventricular function. At 12 weeks there was a non-significant trend towards increasing ventricular wall thickness. However, by 24 weeks, dox treated transgenic animals (Tg +) exhibited ventricular and posterior wall thickening, suggesting hypertrophic cardiomyopathy when compared to other control groups ($p < 0.05$, two-way ANOVA; *Figure 3d,f,g*). Together, these observations indicate that the transgenic mice exhibit progressive cardiomyopathy due to reduced level of *Fxn*, supporting the utilization of Tg + mice for examining the molecular mechanisms downstream of *Fxn* deficiency responsible for cardiac defects.

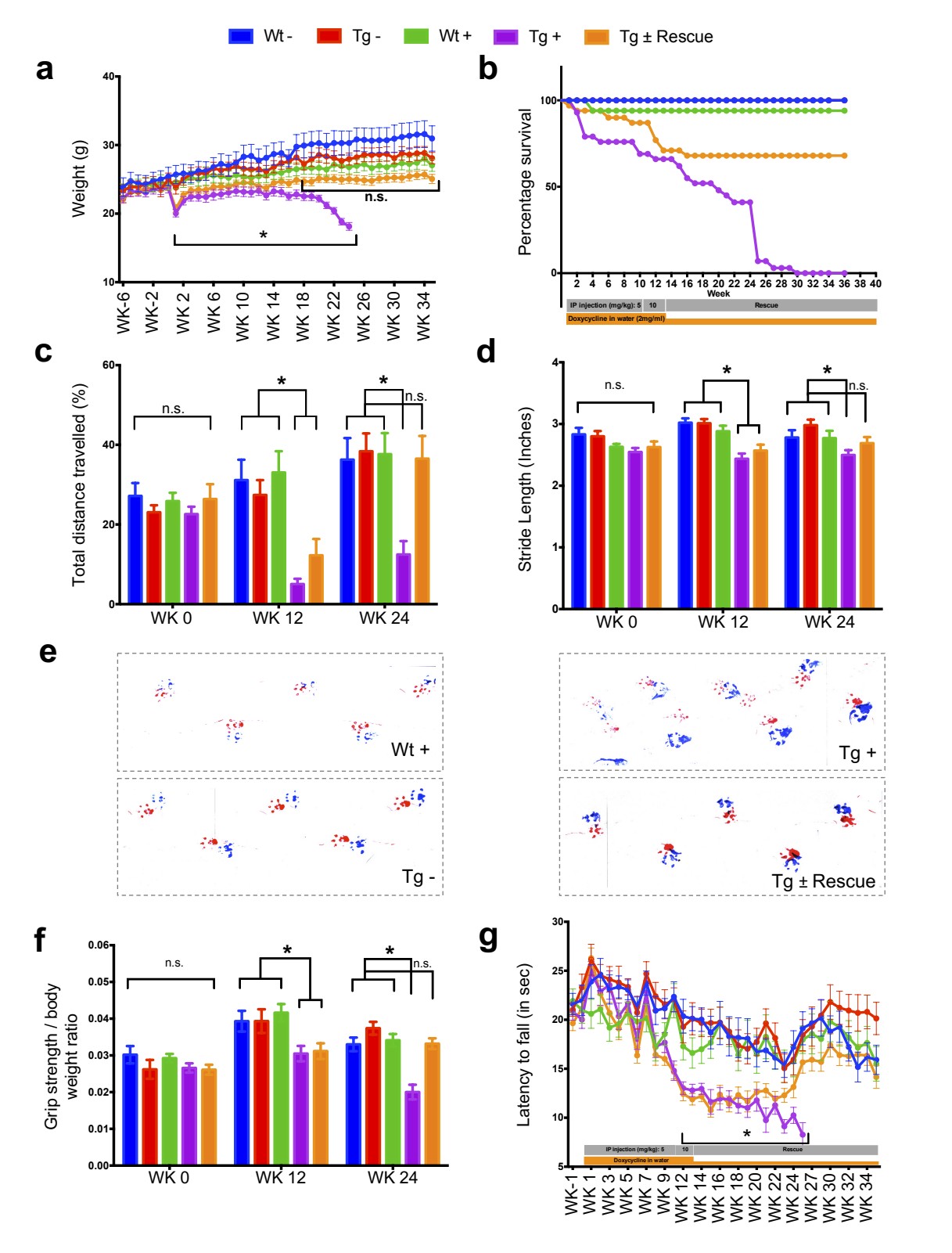

**Figure 2.** Neurological deficits due to frataxin knockdown. Body weight, survival, open field, gait analysis, grip strength and Rotarod in five groups of animals; wild-type mice with dox (Wt +, n = 16 (WK 0), n = 16 (WK 12), n = 16 (WK 24)) and without dox (Wt -, n = 16 (WK 0), n = 16 (WK 12), n = 16 (WK 24)), transgenic mice with dox (Tg +, n = 30 (WK 0), n = 21 (WK 12), n = 15 (WK 24)) and without dox (Tg -, n = 16 (WK 0), n = 15 (WK 12), n = 15 (WK 24)), and transgenic mice with dox removal (Tg ± Rescue, n = 30 (WK 0), n = 21 (WK 12), n = 20 (WK 24)). (a) Body weight from before 6 weeks and upto
*Figure 2 continued on next page*

*Figure 2 continued*
34 weeks after dox treatment. (b) Survival was significantly diminished in dox treated Tg + animals, no mortality was observed after dox withdrawal (Tg ± Rescue). (c) Open field test showing significant decline in total distance traveled by the dox treatment transgenic animals (Tg +) at 12 and 24 weeks when compared across all other groups. After dox withdrawal, there were no differences between the rescue group (Tg ± Rescue) and the three control groups at week 24. (d) Gait footprint analysis of all five groups of mice at 0, 12, and 24 weeks was evaluated for stride length. Dox treated transgenic (Tg +) animals revealed abnormalities in walking patterns displaying significantly reduced stride length; however, the rescue group (Tg ± Rescue) displayed normal stride length when compared to other groups. (e) Representative walking footprint patterns. (f) Grip-strength test, dox treated transgenic (Tg +) mice had reduced forelimb grip strength at 12 and 24 weeks when compared across all other groups. After dox withdrawal, there were no significant differences between the rescue group (Tg ± Rescue) and the three control groups at week 24. (g) Rotarod test in mice upto 34 weeks after dox treatment. Dox treated transgenic (Tg +) animals stayed less time on the Rotarod than the control groups, although after dox withdrawal, there was no significant difference between the rescue group (Tg ± Rescue) and the three control groups. Values between all five groups are shown as mean ±SME. Two-way ANOVA test *$p \leq 0.001$; n.s., not significant.
DOI: https://doi.org/10.7554/eLife.30054.006
The following source data and figure supplements are available for figure 2:

**Source data 1.** This spreadsheet contains the raw data which was used to generate the graphs shown in *Figure 2* after frataxin knockdown during various behavioral tests in FRDAkd and control animals.
DOI: https://doi.org/10.7554/eLife.30054.009
**Figure supplement 1.** Gait analysis measurements reveals decreased stride length in *Fxn* knockdown animals.
DOI: https://doi.org/10.7554/eLife.30054.007
**Figure supplement 2.** Behavioral changes at twelve weeks in FRDAkd mice.
DOI: https://doi.org/10.7554/eLife.30054.008

## Cardiac pathology observed with frataxin knockdown

We next explored the pathological consequences of FRDA knockdown in FRDAkd mice. In FRDA patients, reduced frataxin induces severe myocardial remodeling, including cardiomyocyte iron accumulation, myocardial fibrosis and myofiber disarray (*Koeppen, 2011*). Indeed, we observed substantially increased myocardial iron in Tg + mice, as evidenced by increased ferric iron staining (*Figure 4a* and *Figure 4—figure supplement 1*) and the increased expression of iron metabolic proteins, ferritin and ferroportin, at 20 weeks (*Figure 4b* and *Figure 4—figure supplement 1*). Cardiac fibrosis is commonly found in association with cardiac hypertrophy and failure (*Conrad et al., 1995*). Histological analysis by Masson's trichrome staining revealed excessive collagen deposition in Tg + mice hearts at 20 weeks when compared to other control groups, suggesting cardiac fibrosis (*Figure 4c*). Further examination of cardiomyocyte ultrastructure by electron microscopy in control mouse (Wt +) heart demonstrates normally shaped mitochondria tightly packed between rows of sarcomeres (*Figure 4d*). In contrast, Tg + mice demonstrate severe disorganization, displaying disordered and irregular sarcomeres with enlarged mitochondria at 20 weeks (*Figure 4d*). In a minority of cases, but never in controls, we observed mitochondria with disorganized cristae and vacuoles in Tg + mouse heart at 20 weeks, suggesting mitochondrial degeneration (*Figure 4e*). Next, by examining aconitase, an Fe-S containing enzyme whose activity is reduced in FRDA patients (*Bradley et al., 2000*; *Rötig et al., 1997*), activities in Tg + and other control groups, we observed decreased aconitase activity in the Tg + mouse heart at 20 weeks. Together these observations suggest that the knockdown of *Fxn* in mice causes cardiac pathology similar to that observed in patients (*Smyth, 2005*).

## Frataxin knockdown causes neuronal degeneration

In FRDA patients and mice with *Fxn* conditional knockout, a cell population that is seriously affected by frataxin reduction is the large sensory neurons of dorsal root ganglia (DRG), which results in their degeneration (*Koeppen, 2011*). As in heart tissue, lack of *Fxn* also induces mitochondrial dysfunction in the DRG (*Rötig et al., 1997*). Thus, we assessed whether mitochondrial abnormalities are observed in Tg + mice DRG neurons by electron microscopy (Materials and methods). As anticipated, no mitochondrial abnormalities were detected in DRG neurons in the control groups, but a significant increase in condensed mitochondria were detected in DRG neurons of Tg + mice at 20 weeks (p<0.05, two-way ANOVA; *Figure 5*), but not in Tg - or Wt + mice. In most case, we observed empty vesicles associated with condensed mitochondria in DRG neurons of Tg + animals, but not in other control groups, suggesting neuronal degeneration, similar to what is observed in FRDA

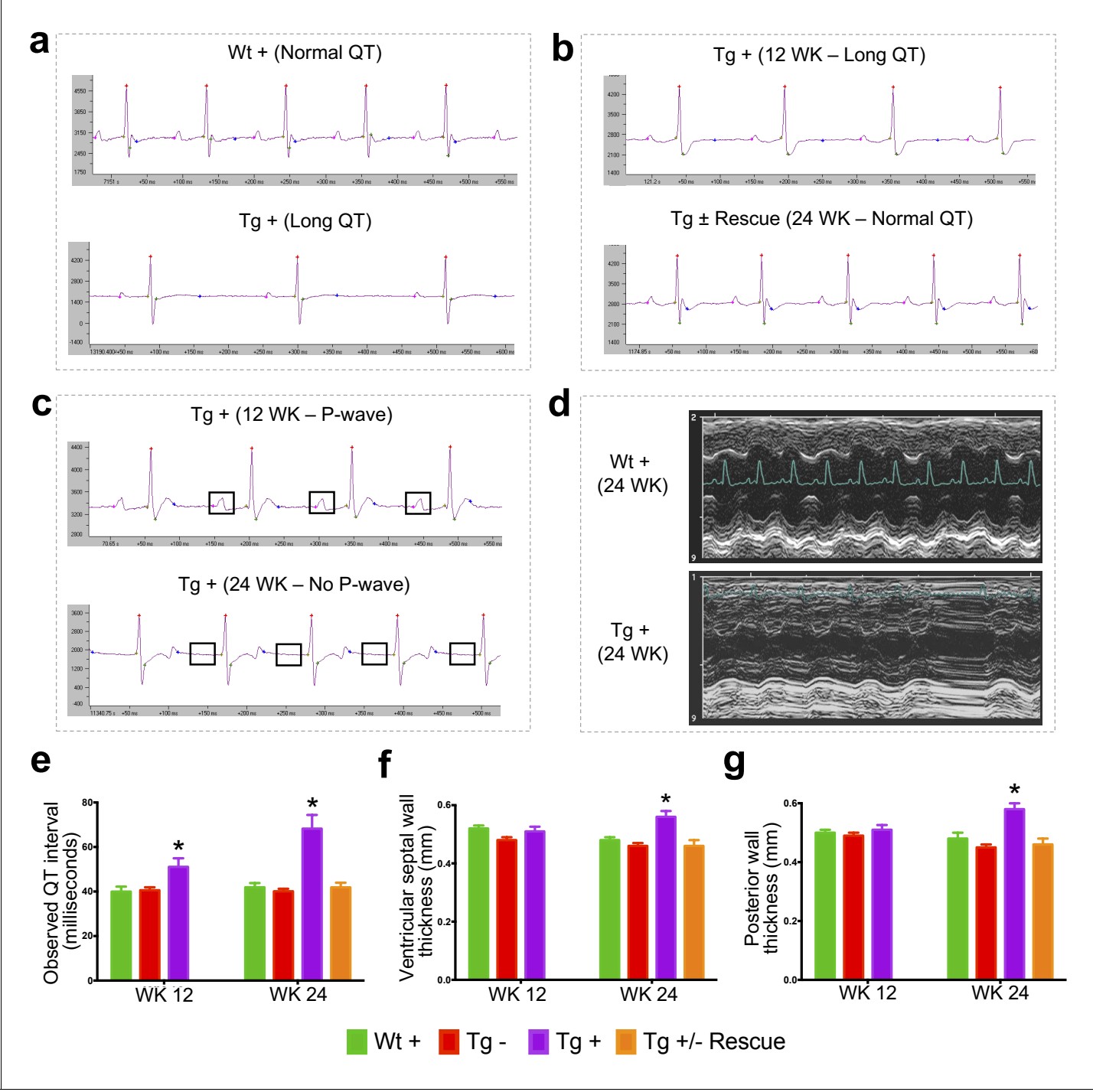

**Figure 3.** Frataxin knockdown mice exhibit signs of cardiomyopathy. (a) Representative traces of ECG recording in a wild-type and transgenic animal after dox treatment for 20 weeks, showing long QT intervals in Tg + animal. (b) ECG recording of a same dox treated transgenic (Tg +) animal at 12 and after dox withdrawal for additional 12 weeks (Tg ± Rescue), showing normal QT interval. (c) ECG recording of the same dox treated transgenic (Tg +) animal at 12 and 24 weeks, showing absence of P-wave only at 24 weeks. (d) Representative echocardiograms from the left ventricle of a wild-type and transgenic mouse at 24 weeks after dox treatment. (e–f) Quantification of observed QT interval (e), ventricular septal wall thickness (f) and posterior wall thickness (g) are shown for weeks 12 and 24. N = 6–8 animals per group, *=p < 0.05; Student's t test. Error bars represent mean ±SEM.

DOI: https://doi.org/10.7554/eLife.30054.010

The following source data is available for figure 3:

**Source data 1.** This spreadsheet contains the electrocardiogram and echocardiogram quantification data after frataxin knockdown during week 12 and 24 in FRDAkd and control animals which was used to generate the bar plots shown in *Figure 3e–g*.

DOI: https://doi.org/10.7554/eLife.30054.011

patients (*Koeppen, 2011*) (p<0.05, two-way ANOVA; Materials and methods; *Figure 5b,c*). Nile Red staining in the DRG neurons of Tg + animals did not display elevated levels of lipid droplets, suggesting further evaluation is needed to identify the composition of these electron-light vesicles (*Figure 5—figure supplement 1*). Previous results in human post-mortem spinal cords of FRDA patients showed a decrease in axon size and myelin fibers (*Koeppen and Mazurkiewicz, 2013*). By analyzing more than 2000 axons per group in the lumbar spinal cord cross-section of high-resolution electron micrographs, we observed significant reduction in axonal size and myelin sheath thickness in the spinal cord samples of Tg + mice when compared to other control groups at 20 weeks (p<0.05, two-way ANOVA; *Figure 6a–c*). In the cerebellum, the number of Purkinje cells and the cerebellar granular and molecular layers were not altered due to *Fxn* knockdown at 20 weeks (*Figure 6—figure supplement 1*).

Next we explored the visual system, because various visual field defects have been reported in patients, suggesting that photoreceptors and the retinal pigment epithelium (RPE) may be affected (*Seyer et al., 2013*; *Fortuna et al., 2009*). By electron microscopic examination of the photoreceptors in the retina, which are specialized neurons capable of phototransduction, we observed disruption in Tg + mice at 20 weeks (Materials and methods; *Figure 6d*). Previous work has shown that the disruption of photoreceptors is related to visual impairment (*Saxena et al., 2014*). Similarly, we also found a significant increase in degenerating RPE cells with vacuoles in Tg + mice, which is involved in light absorption and maintenance of visual function (*Figure 6e*). Together, these results suggest that *Fxn* knockdown in the CNS of adult mice results in degeneration of retinal neurons.

## The *Pdk1/Mef2* pathway and reactive oxygen species in FRDAkd mice

Recently, *Pdk1/Mef2* pathway activation has been associated with iron toxicity due to FXN loss in the neonatal mouse brain and fly (*Chen et al., 2016a*; *Chen et al., 2016b*). To examine this in the FRDAkd model, we performed Western blot analyses to evaluate the phosphorylated levels of PDK and performed RT-PCR analyses to measure the top five candidate target genes of *Mef2* observed in Chen et al. (*Chen et al., 2016b*) (*Figure 6—figure supplement 2*). We measured pPDK levels in brain, muscle, heart, and liver samples after chronic adult *Fxn* knockdown, but did not observe changes in the phosphorylation of S241 in the PDK1 activation loop, which is required for its activity (*Figure 6—figure supplement 2*). Next, we analyzed the mRNA levels of the top five candidate target genes of *Mef2* (*Sgca, Hrc, Nr4a1, Myom1,* and *Tcap*) in cerebellum and heart after *Fxn* knockdown (*Chen et al., 2016b*). In two independent experiments, each utilizing four biological replicates, we found that *Sgca* (one of five genes tested) was significantly over-expressed after *Fxn* knockdown in the cerebellum. In the heart, we found *Nr4a1* was significantly over-expressed, while two of the other *Mef2c* targets, *Hrc* and *Tcap,* were significantly down-regulated after *Fxn* knockdown. These results suggest that the *Pdk1/Mef2* pathway is not consistently or universally activated in mice with adult FRDAkd following iron accumulation and also suggest tissue-specific pathway activation due to *Fxn* knockdown (*Figure 6—figure supplement 2*).

There is debate as to whether the pathogenesis of several neurodegenerative diseases, including FRDA, may involve reactive oxygen species (ROS) associated with mitochondrial dysfunction (*Seznec et al., 2005*). Indeed, in FRDA the evidence for elevation of ROS in disease pathogenesis is quite variable (*Chen et al., 2016a*; *Chen et al., 2016b*; *Seznec et al., 2005*; *Tamarit et al., 2016*; *Schulz et al., 2000*; *Shidara and Hollenbeck, 2010*). To address this in our adult onset model, we first measured the levels of two markers of oxidative stress, 3-nitrotyrosine (3NT) and 4-hydroxy-2-nonenal (4-HNE), on brain, liver, and muscle samples from the Wt + and Tg + mice at 12 and 20 weeks, and did not observe any changes in these markers. Next, we utilized the Oxyblot protein oxidation detection kit to detect the carbonyl groups introduced into proteins by oxidative stress, and again found no evidence for increases in levels of 2,4-dinitrophenylhydrazone (DNP-hydrazone). These finding provide evidence that *Fxn* knockdown does not substantially or chronically elevate ROS in adult mice (*Figure 6—figure supplement 3*).

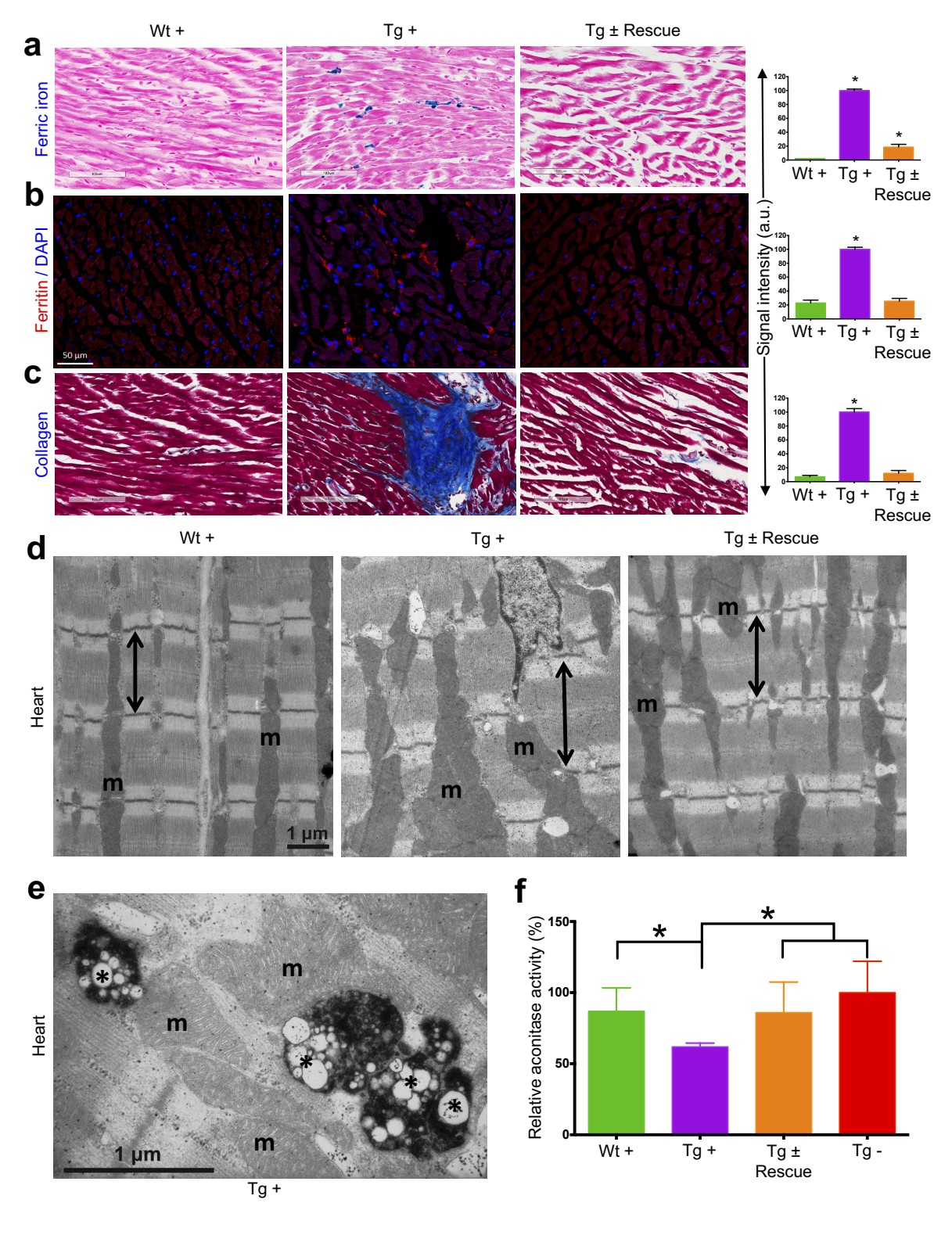

**Figure 4.** Cardiopathology of frataxin knockdown mice. (a) Gomori's iron staining and quantification of iron deposition in dox treated transgenic (Tg +), wild-type (Wt +) and dox withdrawn transgenic (Tg ± Rescue) animals. Dox treated transgenic (Tg +) mice showing myocardial iron-overload (a) also displayed altered expression of ferritin protein (b) which is involved in iron storage. Both iron-overload and ferritin protein levels were significantly lower in Tg ± Rescue animals (a–b). (c) Masson's trichrome staining and quantification showing increased fibrosis in Tg + mice when compared to Wt + and

*Figure 4 continued on next page*

*Figure 4 continued*

Tg ± Rescue animals. (d) Electron micrographs of cardiac muscle from Wt +, Tg + and Tg ± Rescue animals at 20 week after dox treatment. Double arrow lines indicate sarcomere. m = mitochondria. Scale bars, 1 μm. Data are representative of three biological replicates per group. (e) Higher magnification of electron micrographs of cardiac muscle from Tg + mice, showing normal (m) and degenerating mitochondria (asterisks). (f) Aconitase activity was assayed in triplicate in tissues removed from three hearts in each group. Values represent mean ±SME. One-way ANOVA test *p≤0.05.
DOI: https://doi.org/10.7554/eLife.30054.012

The following source data and figure supplement are available for figure 4:

**Source data 1.** This spreadsheet contains the raw signal intensity quantification data of ferric iron, ferritin and collagen staining which was used to generate the graphs shown in *Figure 4a–c* and the aconitase and citrate synthase enzymatic activity measurements are provided (*Figure 4f*).
DOI: https://doi.org/10.7554/eLife.30054.014

**Figure supplement 1.** Frataxin deficiency in mouse heart results in iron accumulation and increased levels of ferritin and ferroportin.
DOI: https://doi.org/10.7554/eLife.30054.013

## Gene expression changes due to frataxin knockdown

Given the phenotypic parallels in the cardiac and nervous system abnormalities in FRDAkd mice with chronic *Fxn* reduction following treatment with dox, we next sought to explore genome wide molecular mechanisms and determine which pathways were affected in the heart and nervous system, and if they were reversible. We analyzed global gene expression profiles in the heart, cerebellum and DRGs from Tg +, Tg -, Wt + and Tg ± mice treated for 0, 3, 12, 16, 20 weeks with dox and a rescue cohort with 4 and 8 weeks post dox treatment (n = 192). Differential expression analyses (Materials and methods) identified 1959 genes differentially expressed in Tg + mice heart when compared to Wt + and Tg - mice (FDR < 5%, *Figure 7a*, *Supplementary file 1*). Similarly, we observed 709 and 206 genes differentially expressed in cerebellum and DRGs of Tg + mice, respectively (*Figure 7b*). Although cross tissue overlap in expression changes was significant, the majority of changes were tissue specific; only 31% and 38% of genes that were differentially expressed in the Tg + mice cerebellum and DRGs were also dysregulated in heart. Likewise, we only observed a 19% overlap between differentially expressed genes in the DRG and cerebellum, consistent with previous observations that *Fxn* reduction causes distinct molecular changes in different tissues (*Coppola et al., 2009*).

Next, we analyzed these differently expressed transcripts in cardiac tissue from Tg + mice with respect to cellular pathways. The top GO categories and KEGG pathways include chemotaxis, immune response, lysosome and phagocytosis, vesicle transport and endocytosis, p53 signaling pathway, cell cycle and division, protein transport and localization, nucleoside and nucleotide binding, and mitochondrion (Benjamini corrected p-value<0.05) (*Figure 7a*, *Supplementary file 2*). To characterize the temporal patterns of these signaling cascades after frataxin knockdown and rescue, we examined their time course by PCA analyses of the gene expression profiles (*Figure 7c*). By examining the cumulative explained variability of the first three principal components for these clusters of genes, we show that each of these functional groups are activated as early as three weeks after dox initiation (and remain elevated for up to 20 weeks); importantly, the aberrant expression of all of these clusters observed in Tg + mice are largely reversed after eight weeks of rescue via *Fxn* re-expression (*Figure 7a,b*).

Another notable observation is that immune system activation is among the earliest pathways regulated after *Fxn* knockdown (*Figure 7c*). This suggests that initiation of immune responses (innate and adaptive) is a direct consequence of *Fxn* knockdown. For example, 38 genes involved in the chemokine signaling pathway (KEGG: mmu04062) were significantly differentially expressed due to *Fxn* knockdown in Tg + mice heart (*Figure 7—figure supplement 1*). Cross validating these genes with previously published gene expression datasets obtained from FRDA patients (*Coppola et al., 2011*) and mouse models associated with FRDA (*Miranda et al., 2002*; *Puccio et al., 2001*), identified several genes involved in the chemokine signaling pathway (E.g.: *Ccl2, 3, 4, 7, Cxcl1, 16, Prkcd, Stat3*) consistently differentially expressed in these six independent FRDA associated datasets (*Figure 7—figure supplement 2*). This implicates a vital role for chemokines and immune response in FRDA pathology, as has been suggested for other neurodegenerative diseases (*Cartier et al., 2005*; *Andreasson et al., 2016*; *Fung et al., 2017*; *Leszek et al., 2016*).

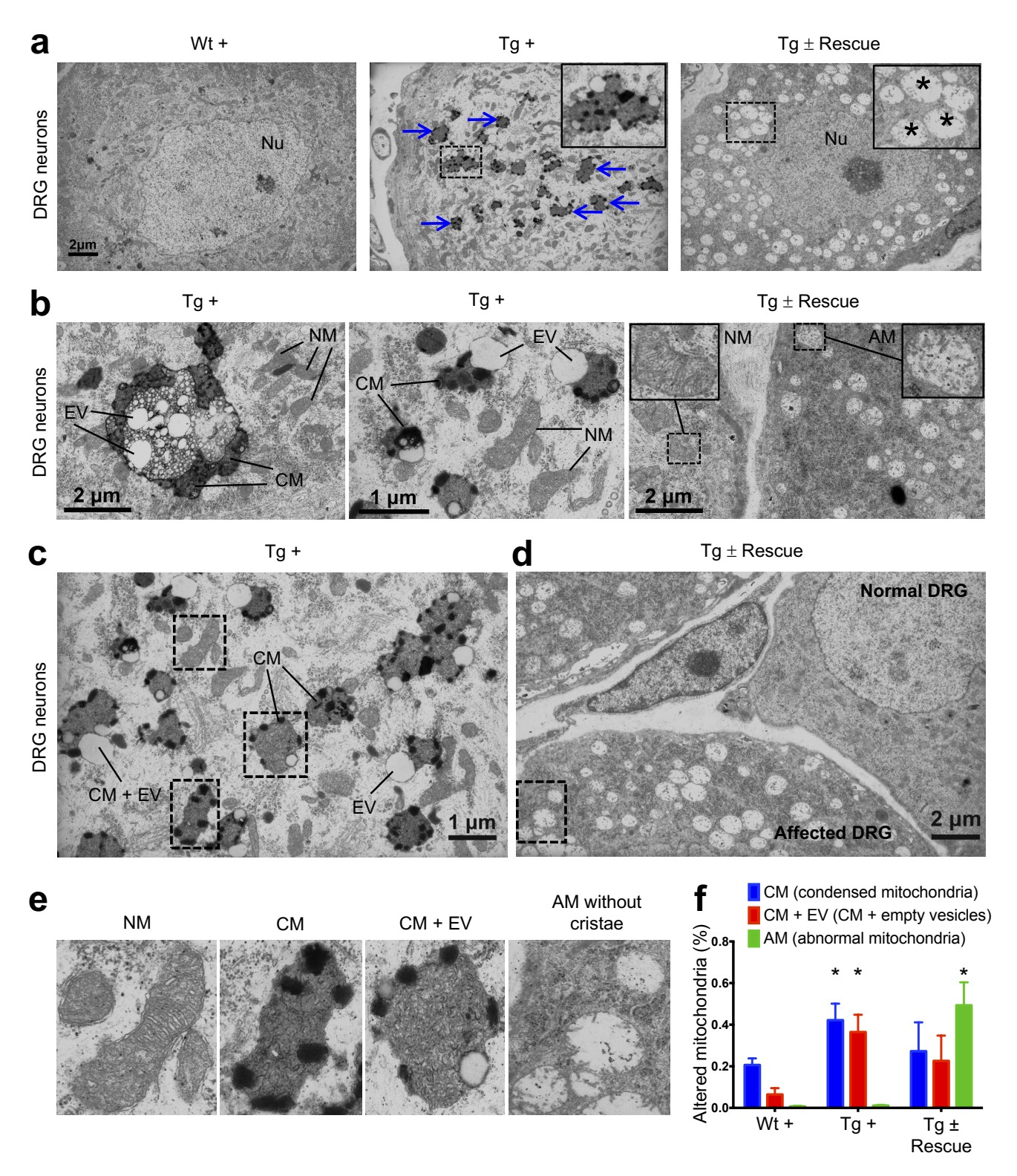

**Figure 5.** Frataxin knockdown mice exhibit neuronal degeneration. (**a**) Electron microscopic analysis of Wt +, Tg + and Tg ± rescue animal DRG neurons at 20 week after dox treatment. Arrows indicate condensed mitochondria with empty vesicles. Insert in Tg + panel shows higher magnification of electron micrographs of condensed mitochondria with empty vesicles in DRG neurons. Insert in Tg ± Rescue panel shows higher magnification of abnormal mitochondria without cristae (asterisks). Nu = nucleus. (**b**) Higher magnification of electron micrographs of Tg + and Tg ± Rescue animal DRG

*Figure 5 continued on next page*

*Figure 5 continued*

neurons at 20 week after dox treatment. Tg + panel shows degenerating mitochondria and condensed mitochondria with empty vesicles in DRG neurons. Tg ± Rescue panel shows two DRG neurons, one consisting of normal mitochondria (insert) and the other neuron with abnormal mitochondria without cristae (insert). (c) Higher magnification of Tg + animals showing condensed mitochondria with empty vesicles in DRG neurons. (d) Higher magnification of Tg ± rescue animals showing two DRG neurons, one consisting of normal mitochondria (normal DRG) and the other neuron with abnormal mitochondria without cristae (affected DRG). (e) Insert images from c and d panels shows higher magnification of normal mitochondria (NM), condensed mitochondria (CM), condensed mitochondria along with empty vesicles (CM + EV), and abnormal mitochondria without cristae (AM). (f) Quantification of altered mitochondria in DRG neurons. Data are from multiple images from three biological replicates per group. Values represent mean ± SME. Two-way ANOVA test *=P $\leq$ 0.05.

DOI: https://doi.org/10.7554/eLife.30054.015

The following source data and figure supplement are available for figure 5:

**Source data 1.** This spreadsheet contains the manual mitochondrial counting data from electron microscopy images of the DRG neurons which was used to generate the graph shown in *Figure 5f* after frataxin knockdown in FRDAkd and control animals.
DOI: https://doi.org/10.7554/eLife.30054.017
**Figure supplement 1.** Nile Red staining labeling lipid droplets.
DOI: https://doi.org/10.7554/eLife.30054.016

## Shared and tissue specific gene expression changes

We next performed weighted gene co-expression network analysis (WGCNA) (*Zhang and Horvath, 2005*; *Langfelder and Horvath, 2008*; *Geschwind and Konopka, 2009*), a powerful method for understanding the modular network structure of the transcriptome, to organize the frataxin related transcriptional changes in an unbiased manner following knockdown and rescue. WGCNA permits identification of modules of highly co-expressed genes whose grouping reflects shared biological functions and key functional pathways, as well as key hub genes within the modules, and has been widely applied to understanding disease related transcriptomes (*Zhang et al., 2013*; *Parikshak et al., 2016*). By applying WGCNA and consensus network analysis (*Langfelder and Horvath, 2008*; *Chandran et al., 2016*), we identified 19 robust, reproducible co-expression modules (Materials and methods; *Figure 7—figure supplement 3* and *Supplementary file 3*) between the three tissues datasets generated at 0, 3, 12, 16 and 20 weeks after *Fxn* knockdown and 4 and 8 weeks post dox removal. On the basis of the module eigengene correlation with time-dependent changes after *Fxn* knockdown, we first classified modules as up-regulated and down-regulated following knockdown (*Figure 7—figure supplement 4*). Next, based on the significant module trait relationships (Wilcoxon p-value<0.05), we identified 11 modules strongly associated with *Fxn* knockdown: three down-regulated modules in two or more tissues after *Fxn* knockdown (yellow, lightgreen and turquoise) and three up-regulated modules (blue, purple, and black) (*Figure 7—figure supplement 4*). There also were three down-regulated modules in heart that were up-regulated in cerebellum (red, greenyellow and magenta) and two up-regulated modules in heart that were down-regulated in cerebellum (cyan and pink). Although six of the gene co-expression modules (yellow, lightgreen, turquoise, blue, purple, and black) in the heart, cerebellum and DRG following *Fxn* knockdown are highly preserved across tissues, five modules (red, greenyellow, magenta, cyan and pink) exhibit differential expression profiles suggesting tissue specific molecular changes, consistent with previous observations of shared and organ specific changes (*Coppola et al., 2009*) (*Figure 7—figure supplement 4*).

As a first step toward functional annotation of the cross-tissue modules, we applied GO and KEGG pathway enrichment analyses, which showed enrichment (Benjamini-corrected p values < 0.05) for several GO categories and pathways in the *Fxn* knockdown co-expression modules which included several previously associated functional categories related to current concepts of frataxin function (*Supplementary file 4*). Three modules (yellow, lightgreen and turquoise) that were down-regulated in two or in all three tissues due to *Fxn* knockdown included, nucleotide, nucleoside and ATP binding, myofibril assembly, muscle tissue development, RNA processing, and several mitochondrial related categories: oxidative phosphorylation, respiratory chain, NADH dehydrogenase activity, and electron transport chain. We also observed that the genes present in turquoise module were enriched for several KEGG pathways, namely, PPAR signaling (mmu03320; genes = 14), insulin signaling (mmu04910; n = 19), fatty acid metabolism (mmu00071; n = 10), cardiac muscle contraction (mmu04260; n = 20), dilated cardiomyopathy (mmu05414; n = 13), and hypertrophic

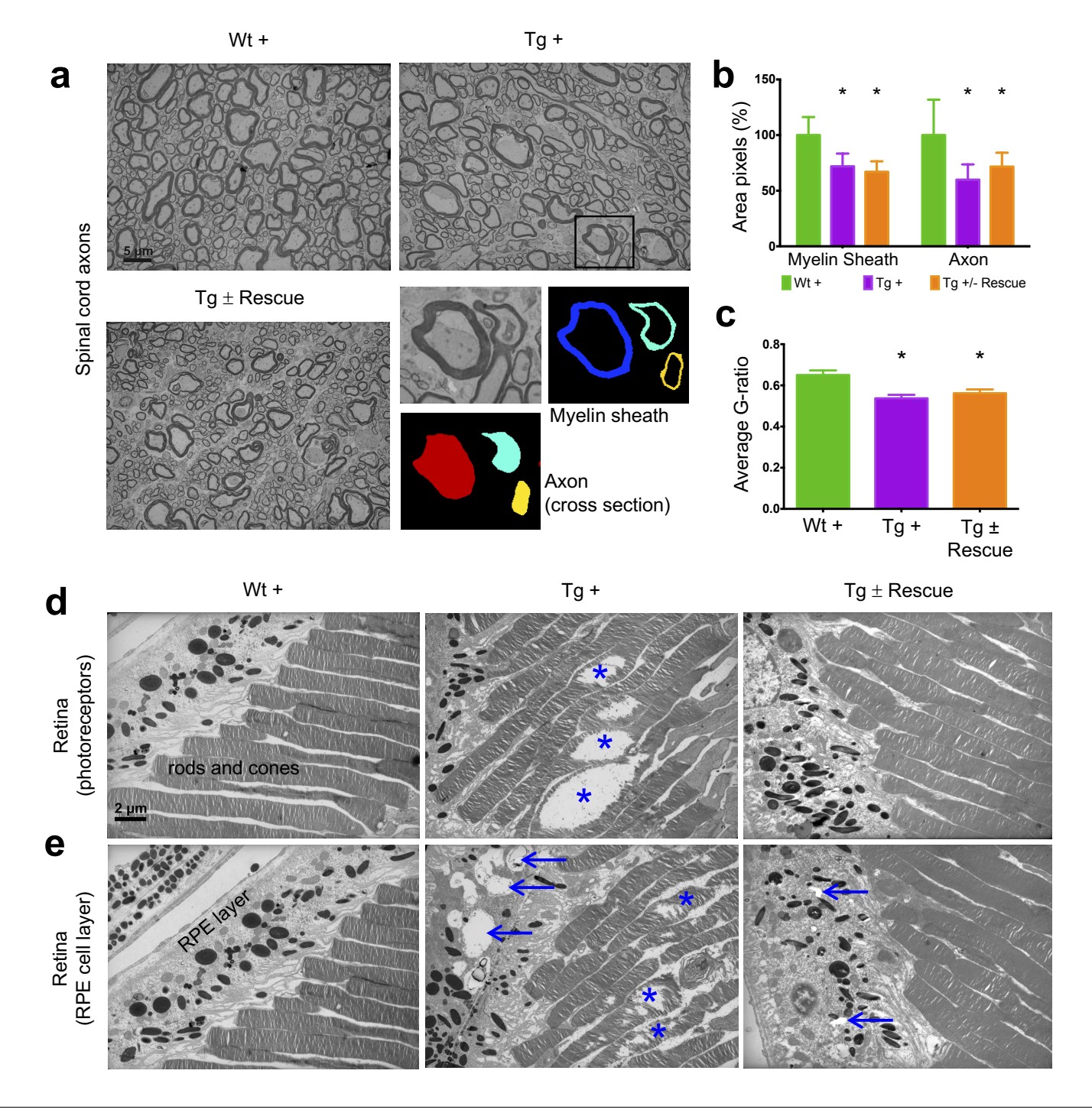

**Figure 6.** Frataxin knockdown mice exhibit neuronal degeneration in the spinal cord and retina. Electron microscopic analysis of Wt +, Tg + and Tg ± rescue animal at 20 week after dox treatment. (**a**) Electron micrographs of spinal cord axon cross-section, displaying reduced myelin sheath thickness and axonal cross-section area in Tg + and Tg ± Rescue animals. Bottom panel shows representative area utilized for quantification. (**b–c**) Quantification of myelin sheath thickness and axonal cross-section area in the spinal cord. Data are from 2000 or more axons per group in the lumbar spinal cord cross-section of high-resolution electron micrographs from three biological replicates per group. Values represent mean ±SME. One-way ANOVA test *=P ≤ 0.05. (**d**) Electron micrographs of rod and cone photoreceptor cells, showing their disruption in Tg + animals (asterisks). (**e**) Retinal pigment epithelium cell layer showing the presences of large vacuoles (arrows) in Tg + animals along with disruption in their photoreceptor cells (asterisks).
DOI: https://doi.org/10.7554/eLife.30054.018

The following source data and figure supplements are available for figure 6:

*Figure 6 continued on next page*

*Figure 6 continued*

**Source data 1.** This spreadsheet contains the image analyses quantification data from electron microscopy images of the spinal cord axons which was used to generate the graph shown in *Figure 6b and c* after frataxin knockdown in FRDAkd and control animals.
DOI: https://doi.org/10.7554/eLife.30054.022
**Figure supplement 1.** Frataxin knockdown in adult mice does not change the numbers of Purkinje cells and thickness of granular layer of the cerebellum.
DOI: https://doi.org/10.7554/eLife.30054.019
**Figure supplement 2.** PDK1 and *Mef3* pathway is not activated in *Fxn* knockdown mice.
DOI: https://doi.org/10.7554/eLife.30054.020
**Figure supplement 3.** Quantification of reactive oxygen species (ROS) levels in *Fxn* knockdown animals.
DOI: https://doi.org/10.7554/eLife.30054.021

cardiomyopathy (mmu05410; n = 14), which have been previously associated with FRDA, reflecting the multi-systemic nature of FRDA. Similarly, three upregulated cross tissue modules (blue, purple, and black) include, nucleotide binding, vesicle-mediated transport, immune response (innate and adaptive), defense response, inflammatory response, induction of apoptosis, positive regulation of cell death, cell adhesion, and skeletal system development (*Supplementary file 4*). These results demonstrate that unsupervised analyses can identify groups of genes not only with shared biological functions, but also relevant to the clinical phenotypes observed in FRDA.

Three tissue specific modules that were down-regulated in heart and up-regulated in cerebellum (red, greenyellow and magenta) showed enrichment for, transcription regulator activity, neurological system process, synaptic vesicle and nucleotide, nucleoside and ATP binding. Two other modules that were up-regulated in heart and down-regulated in cerebellum (cyan and pink) were enriched for cell cycle, cell division, mitosis and DNA replication (*Supplementary file 4*). In summary, we observed several metabolic functional categories that were differentially expressed (up and down) due to *Fxn* knockdown. The modules consisting of mitochondrial and cardiac specific categories along with PPAR signaling, insulin signaling, fatty acid metabolism pathways were down regulated in all tissues. Likewise, the modules enriched for immune, apoptosis and cell death related categories we up-regulated in all tissues due to *Fxn* knockdown. Synaptic and transcription regulator activity functional categories were only up-regulated in cerebellum, whereas cell cycle, cell division, mitosis and DNA replication related functional categories were down-regulated in cerebellum and up-regulated in the heart. These general functional categories related to *Fxn* knockdown have been previously associated with altered function in FRDA patients (*Coppola et al., 2011*; *Haugen et al., 2010*), suggesting that genes within these modules would make interesting candidate genes for follow up studies, because many of the genes have not been previously associated with FRDA pathology and the disease mechanism.

## Gene expression candidate biomarkers associated with frataxin knockdown

To identify candidate molecular targets and to better understand the molecular pathophysiology associated with *Fxn* knockdown, we first manually combined all the GO ontology terms (see above and *Supplementary file 4*) that were enriched in the 11 modules into 26 broad functional categories based on GO slim hierarchy (*Supplementary file 5*) and screened for co-expressed genes within each functional category in all three tissues (r > 0.5 and p-value<0.05) over the time course. This allowed us to identify critical functional sub-categories that are up or down regulated due to frataxin knockdown and subsequently permitted us to detect differentially expressed candidate genes that are co-expressed within each functional category (*Figure 7d*; *Figure 7—figure supplement 5*). For example, we show that immune, cell cycle and apoptosis related functional groups are up-regulated, whereas cardiac and mitochondrial related functional groups were down-regulated (*Figure 7d*). In the immune category, we observed most prominent changes in complement activation pathway genes, namely, *C3, C4b, C1qb, C1qc and Serping1*. Interesting, we also observed that many of these genes were also up-regulated in peripheral blood mononuclear cells obtained from FRDA patients (*Figure 7—figure supplement 6*), suggesting the potential for complement activation to act as a biomarker for FRDA as previously suggested for other degenerative diseases (*Aiyaz et al., 2012*). Similarly, we found several genes, for example, *Cacna2d1, Abcc9 and Hrc* involved in normal

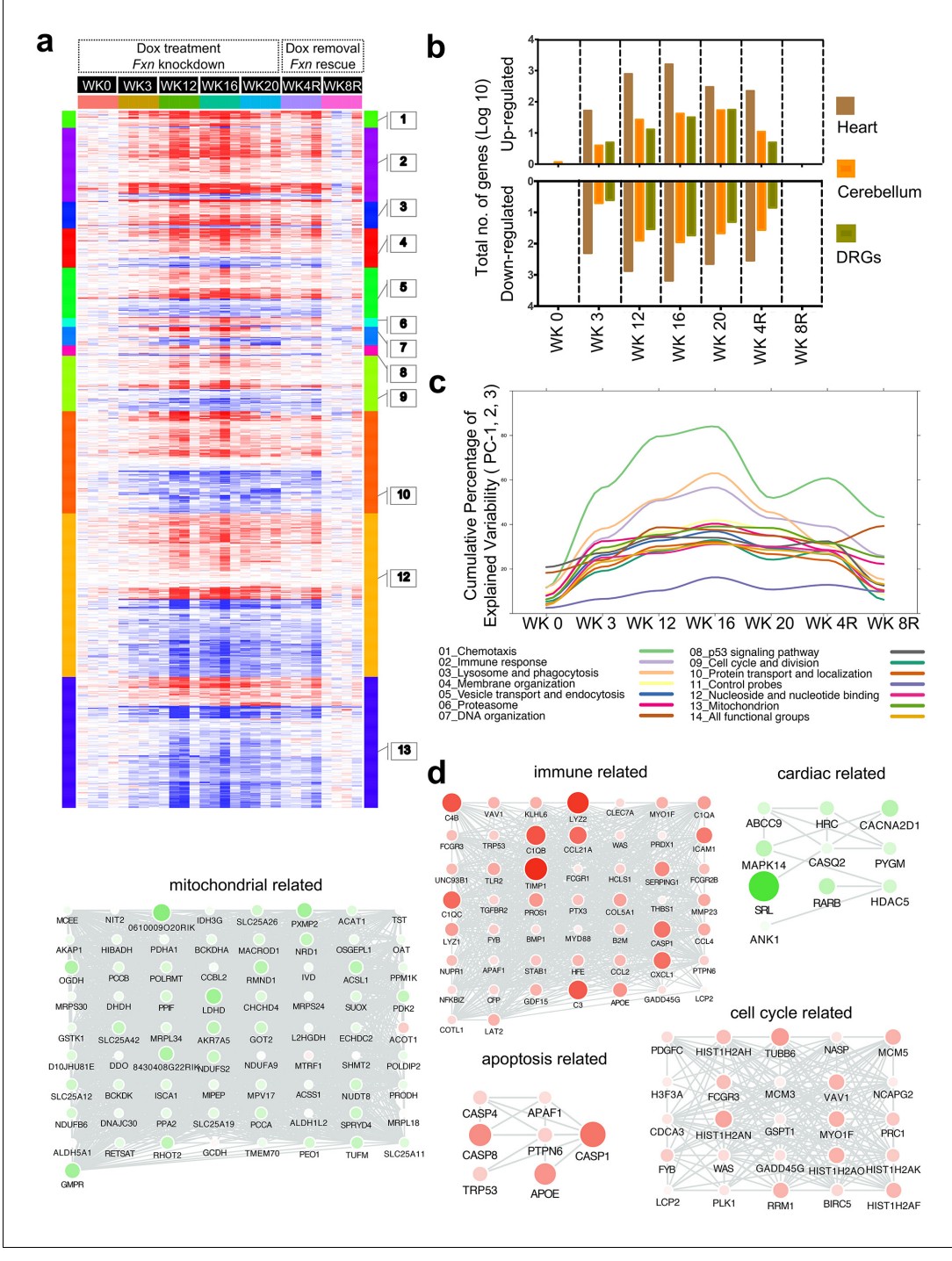

**Figure 7.** Gene expression analysis of frataxin knockdown mice. (a) Heat map of significantly up- and down-regulated genes (rows) in heart tissue of Tg + mice from 0, 3, 12, 16, 20 and plus 4, 8 weeks post dox treatment relative to controls are grouped into 13 functional categories. (b) Summary of differentially expressed genes during *Fxn* knockdown and rescue in heart, cerebellum and DRG tissues from four biological replicates. (c) Cumulative percent of variability in Tg + gene expression data explained by the first three principal component for each functional category. (d) Networks highlighting differentially expressed genes due to *Fxn* knockdown in Tg + mice for selected functional categories. Nodes represents genes and edges are present between nodes when their gene expression correlation is greater than 0.5. Mouse gene names are displayed in upper case for clarity purpose. Node size and color (red = up regulation and green = down regulation) denotes extent of differential expression.

*Figure 7 continued on next page*

*Figure 7 continued*

DOI: https://doi.org/10.7554/eLife.30054.023

The following source data and figure supplements are available for figure 7:

**Source data 1.** This spreadsheet contains the number of genes differentially expressed in the microarray data from heart, cerebellum and DRGs after frataxin knockdown in FRDAkd and control animals (*Figure 7b*) and the cumulative percent of variability data from PCA analyses is also provided which was used to generate the graph shown in *Figure 7c*.
DOI: https://doi.org/10.7554/eLife.30054.030
**Figure supplement 1.** Chemokine signaling pathway is altered in frataxin knockdown mice.
DOI: https://doi.org/10.7554/eLife.30054.024
**Figure supplement 2.** Chemokine signaling pathway is altered in FRDA patients and mouse models.
DOI: https://doi.org/10.7554/eLife.30054.025
**Figure supplement 3.** Identification of frataxin knockdown specific modules using WGCNA.
DOI: https://doi.org/10.7554/eLife.30054.026
**Figure supplement 4.** WGCNA identifies consensus co-expression modules associated with frataxin knockdown and rescue.
DOI: https://doi.org/10.7554/eLife.30054.027
**Figure supplement 5.** Co-expression analyses reveals functional categories associated with frataxin knockdown and rescue.
DOI: https://doi.org/10.7554/eLife.30054.028
**Figure supplement 6.** Frataxin knockdown alters complement activation pathway genes in adult mice.
DOI: https://doi.org/10.7554/eLife.30054.029

cardiac function, to be down-regulated in heart tissue upon frataxin knockdown (*Figure 7d*). *CAC-NA2D1* is associated with Brugada syndrome, also known as sudden unexpected nocturnal death syndrome, a heart condition that causes ventricular arrhythmia (*Risgaard et al., 2013*). Mutations in *ABCC9* gene can cause dilated cardiomyopathy (*Bienengraeber et al., 2004*) and a genetic variant in the *HRC* gene has been linked to ventricular arrhythmia and sudden death in dilated cardiomyopathy (*Singh et al., 2013*). These observations suggest that lower levels of frataxin causes dysregulation of multiple genes related to arrhythmia or cardiac failure, a primary cause of death in FRDA patients.

There has been accumulating evidence suggesting that apoptosis may be an important mode of cell death during cardiac failure (*González et al., 2003*). In agreement with this, we observed genes related to apoptosis were up-regulated after *Fxn* knockdown in FRDAkd mice heart (*Figure 7d*), which has been previously associated with FRDA pathogenesis and reported in other *Fxn* deficiency models (*Simon et al., 2004*; *Huang et al., 2009*; *Bolinches-Amorós et al., 2014*). In order to validate our network findings, we tested CASP8 protein levels (*Muzio et al., 1996*), observing an increase in cleaved Caspase eight protein levels in Tg + heart tissue compared with control mice (*Figure 8a*). Next, we employed the TUNEL assay to detect apoptotic cells that undergo extensive DNA degradation during the late stages of apoptosis (*Kyrylkova et al., 2012*). However, we did not observe an increase in cell death in all tissues by TUNEL staining (*Figure 8b*).

## Literature data extraction for candidate genes associated with frataxin knockdown

We next examined the phenotype-gene associations extracted by co-occurrence-based text-mining in an attempt to link FRDA disease phenotypes with genes. For this, we screened the literature for potential co-occurrence link association between the observed FRDAkd mice phenotypes and the genes that are differentially expressed after *Fxn* knockdown (Materials and methods). Identifying potential biomarker candidates that are previously validated for certain phenotypes can provide insight into disease progression, pathogenesis and extremely valuable for assessing therapeutic options (*Trugenberger et al., 2013*). We screened with the genes that are differentially expressed (FDR < 5%) and present in the co-expression modules associated with behavioral and pathological key-terms (Eg: ataxia; *Supplementary file 6*) in the published literature. Interestingly, this analysis identified numerous genes in which mutations are known to cause Mendelian forms of ataxia namely, *ANO10* (*Vermeer et al., 2010*; *Mišković et al., 2016*), *CABC1* (*Mollet et al., 2008*; *Gerards et al.,*

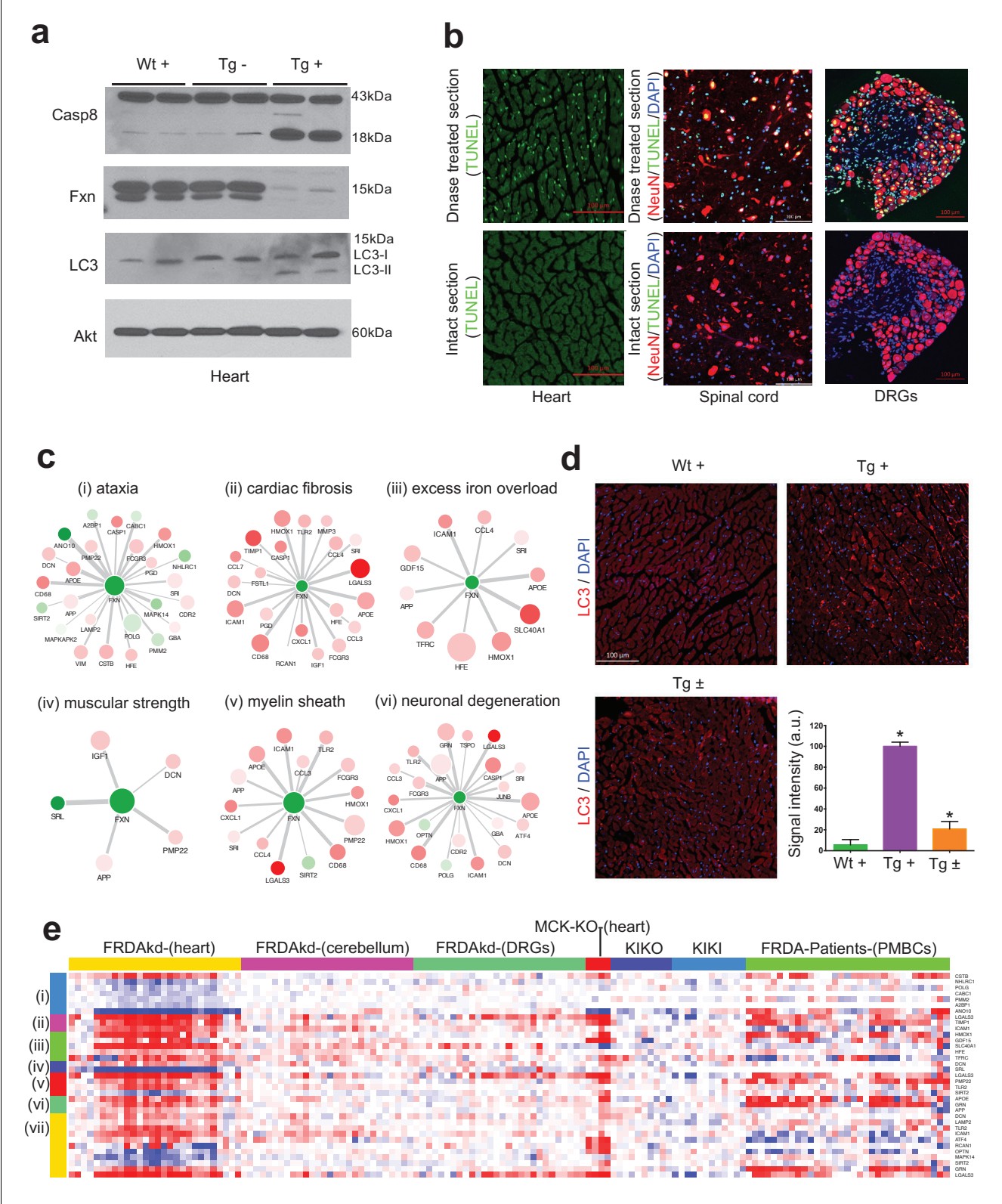

**Figure 8.** Gene expression candidate biomarkers in frataxin knockdown mice. (**a**) Western blot analyses of Caspase 8, FXN and LC3 following 20 weeks with or without dox treatment in heart. (**b**) TUNEL assay of heart, spinal cord and DRG neurons at 20 weeks after dox treatment in Tg + mice. (**c**) Literature associated gene networks highlighting differentially expressed genes due to *Fxn* knockdown in Tg + mice for selected key-terms. Nodes represents genes which has pairwise correlation greater than 0.5 with *Fxn* gene and edge size represents strength of their gene expression correlation.

*Figure 8 continued on next page*

*Figure 8 continued*

Node size correspond to number of PubMed hits with co-occurrence of gene and its corresponding key-term. Node color represents up-regulated (red) and down-regulated (green). Gene names are displayed in upper case for clarity purpose. (**d**) Representative images and quantification of LC3 staining in heart tissue at 20 weeks after dox treatment. Values represent mean from three biological replicates per group ±SME. One-way ANOVA test *=P ≤ 0.05. (**e**) Heat map depicting expression of key genes (rows) across samples (columns) for seven groups (red corresponds to gene up-regulation and blue to down-regulation). The seven groups represents: (i) ataxia, (ii) cardiac fibrosis, (iii) excess iron overload, (iv) muscular strength, (v) myelin sheath, (vi) neuronal degeneration and (vii) autophagy related genes. The column represents seven independent datasets obtained from, FRDAsh mice, cardiac specific knockout mice (*Puccio et al., 2001*), knock-in knockout mice (*Miranda et al., 2002*), knock-in mice (*Miranda et al., 2002*) and patient peripheral blood mononuclear cells (*Coppola et al., 2011*).

DOI: https://doi.org/10.7554/eLife.30054.031

The following source data and figure supplement are available for figure 8:

**Source data 1.** This spreadsheet contains the raw signal intensity quantification data of LC3 staining which was used to generate the graph shown in *Figure 8d*.

DOI: https://doi.org/10.7554/eLife.30054.033

**Figure supplement 1.** Literature mining identifies genes associated with frataxin knockdown and observed phenotype in FRDAkd mice.

DOI: https://doi.org/10.7554/eLife.30054.032

*2010*) and *POLG* (*Schicks et al., 2010*; *Synofzik et al., 2012*) that were significantly down-regulated in the heart tissue and slightly down-regulated in the cerebellum after *Fxn* knockdown in Tg + animals (*Figure 8c* and *Figure 8—figure supplement 1*). We also found three other genes that are differentially expressed due to *Fxn* knockdown, namely, *CSTB* (*Pennacchio et al., 1998*), *NHLRC1* (*Singh and Ganesh, 2009*) and *PMM2* (*Matthijs et al., 1997*) all of which are associated with other disorders manifesting ataxia (*Figure 8c*). These observations suggest that *Fxn* along with multiple other downstream candidates causes behavioral deficits in FRDAkd mice. Similarly, for cardiac phenotypes, we identified multiple genes related to cardiac fibrosis that were up-regulated in FRDAkd mice heart (*Figure 8c*), including *Lgals3* (*Sygitowicz et al., 2016*), *Icam1* (*Salvador et al., 2016*) and *Timp1* (*Polyakova et al., 2011*)(*Figure 8c*). Genes related to iron regulation, included *Hfe* (*Del-Castillo-Rueda et al., 2012*), *Slc40a1* (*Del-Castillo-Rueda et al., 2012*), *Hmox1* (*Song et al., 2012*), *Tfrc* (*Del-Castillo-Rueda et al., 2012*) and *Gdf15* (*Cui et al., 2014*), all of which are directly involved in hemochromatosis and iron overload (*Figure 8c*). We also found previously associated genes related with muscle strength (E.g.: *Srl*, *Dcn*), myelination (E.g.: *Pmp22*, *Lgals3*, *Tlr2* and *Sirt2*) and several genes related to neuronal degeneration (E.g.: *Grn* and *App*) to be dysregulated in FRDAkd mice (*Figure 8c* and *Figure 8—figure supplement 1*), connecting this degenerative disorder with the molecular signaling pathways known to be causally involved in other such disorders.

One pathway that was dysregulated and increasingly associated with neurodegeneration, was autophagy, since several autophagy-related genes (*Lamp2, Atf4, Tlr2, Optn, Mapk14, Sirt2, Icam1, Lgals3, Dcn, Rcan1, Grn*) were present in these disease phenotype-associated sub networks (*Figure 8c*). Autophagy is responsible for the recycling of long-lived and damaged organelles by lysosomal degradation (*Gustafsson and Gottlieb, 2009*) and is associated with various stress conditions including mitochondrial dysfunction (*Pavel and Rubinsztein, 2017*; *Bento et al., 2016*). Disruption of autophagy is also reported as altered in other *Fxn* deficiency models (*Simon et al., 2004*; *Huang et al., 2009*; *Bolinches-Amorós et al., 2014*). To validate our network findings, we utilized LC3-II as a marker for autophagy, showing autophagy activation in Tg + mice heart, but not in spinal cord (*Figure 8a,d*). This suggests activation of apoptosis (*Figures 7d* and *8a*) and autophagy (*Figure 8a,d*) may therefore potentiate the cardiac dysfunction of Tg + mice. Next, by examining the expression levels of all these sub-network genes (*Figure 8c*) in the datasets of other FRDA related mouse models (*Miranda et al., 2002*; *Puccio et al., 2001*) and patient peripheral blood mononuclear cells (*Coppola et al., 2011*), we show that they are also differentially expressed in the same direction in majority of the samples (*Figure 8e*). In contrast, in two asymptomatic mouse models (KIKO and KIKI) with frataxin reduction below the threshold necessary to produce phenotype (*Miranda et al., 2002*), most of the gene expression changes observed here in this symptomatic model are not recapitulated (*Figure 8e*). This suggests that these sub network genes may be strong candidates for molecular biomarkers in FRDA. In summary, consistent with our behavioral, physiological and pathological findings, we show multiple candidate genes related to key degeneration related phenotypes to be altered in FRDAkd mice.

## Rescue of behavioral, pathological and molecular changes due to frataxin restoration

Our data show that mice carrying the dox inducible shRNA for *Fxn* develop normally until they are challenged with dox, which subsequently results in substantial FXN reduction and causes multiple behavioral and pathological features observed in patients with FRDA, including cardiac and nervous system dysfunction (*Figures 2–6*). *Fxn* knockdown mice displayed remarkable parallels with many of the behavioral, cardiac, nervous system impairments along with physiological, pathological and molecular deficits observed in patients. At a general level, FRDAkd mice displayed weight loss, reduced locomotor activity, reduced strength, ataxia and early mortality (*Figure 2*). In the nervous system, FRDAkd mice showed abnormal mitochondria and vacuolization in DRGs, retinal neuron degeneration, and reduced axonal size and myelin sheath thickness in the spinal cord (*Koeppen and Mazurkiewicz, 2013*; *Koeppen, 2011*; *Simon et al., 2004*; *Perdomini et al., 2013*) (*Figures 5* and *6*). With regards to cardiac dysfunction, FRDAkd mice exhibited conduction defects, cardiomyopathy, evidence of iron overload, fibrosis, and biochemical abnormalities that are commonly observed in patients ( (*Koeppen and Mazurkiewicz, 2013*; *Koeppen, 2011*; *Perdomini et al., 2013*) *Figures 3* and *4*). These features correspond to a phenotype of substantial multisystem, clinical disability consistent with moderate to severe disease after three months of frataxin deficiency and that lead to a 50% mortality rate at approximately five months in these mice.

Given that optimal therapy in patients with FRDA could be considered replenishment of FXN itself, e.g. via gene therapy (*Perdomini et al., 2014*) or improvement of *Fxn* transcription via small molecules (*Gottesfeld et al., 2013*), we asked how much of the behavioral, physiological, pathological and molecular phenotype(s) observed at this relatively severe stage of illness could be reversed following such 'optimal' therapy. In this case, optimal therapy is return of normal FXN levels under endogenous regulation through relief of exogenous inhibition. Answering this question of reversibility is crucial for any clinical trial, whatever the mechanism of action of the therapy, since we currently have limited information as to what represents potentially reversible neurologic or cardiac phenotypes.

To address this, we compared two groups of mice, both transgenic and treated with dox for 12 weeks, at which time both groups show equivalent levels of substantial clinical features (*Figure 2— figure supplement 2*). In one, Tg + the dox is continued for another 12 or more weeks, and in the other Tg ± the dox is removed and the animals are followed. Restoration of FXN expression in *Fxn* knockdown mice (Tg ±) that had reached a level of substantial clinical dysfunction led to significant improvement in lifespan (no death until 20 weeks post dox removal) when compared to Tg + group, which resulted in a 90% mortality rate by 25 weeks after dox treatment initiation (*Figure 2b*). Rescue animals (Tg ±) also displayed rapid improvement in: gait ataxia, body weight, muscle strength, locomotor activity, and balance on rotarod test over the ensuing 12 week period, to a point where the treated animals were not significantly different from controls on many tests (*Figure 2*). Remarkably, we observed all of the six FRDA associated clinical phenotypes tested showed significant improvement, suggesting that FRDA-like neurological defects due to absence of the mouse *Fxn* gene can be rectified by delayed restoration of *Fxn* (*Figure 2*).

We next sought to determine whether the observed reversible behavioral changes in FRDAkd mice are also accompanied by recovery of the physiological phenotype in FRDAkd mice heart, since, changes in physiology offer attractive therapeutic targets for symptomatic and preventive treatment of ataxia. Tg ± mice that received the dox for 12 weeks followed by 12 weeks dox removal displayed reversal of the long QT interval phenotype, when compared to Tg + mice at both 12 and 24 weeks post dox treatment initiation (*Figure 3a,b,e*). We observed ventricular and posterior wall thickening only at 24 weeks post dox treatment in Tg + animals (*Figure 3d,f,g*), suggesting that long QT interval phenotype is a earlier manifestation of disease that occurs before left ventricular wall thickening. Correcting this aberrant physiology through activation of *Fxn* gene expression is a potential early therapeutic biomarker.

One question that intrigued us because of the striking behavioral and physiological functional recovery is to what extent frataxin deficiency-associated phenotypes represented pathological findings related to cell dysfunction (potentially reversible) versus cell death (irreversible) recovery. Pathological and biochemical analyses in Tg ± mice heart following 8 weeks of dox withdrawal revealed improved cardiac function and reduced iron and ferritin accumulation, myocardial fibrosis, well-

ordered sarcomeres, normal aconitase activity and reduced mitochondrial abnormalities (*Figure 4*). Relevant to the pathogenesis of FRDA heart and the role of iron and mitochondrial defect, it has been found that cells with these defects are sensitized to cellular dysfunction (*Delatycki et al., 1999*; *Michael et al., 2006*), and here we show this can be ameliorated by *Fxn* restoration.

In the nervous system of Tg ± mice 8 weeks after dox removal, we observed reduced empty vesicles and fewer condensed, degenerating mitochondria in DRG neurons along with several abnormal mitochondria (empty and without cristae) containing DRG neurons compared with those in which dox was continued (*Figure 5a–f*). We only observed mild improvement in myelin sheath thickness and cross section axonal size in the spinal cord of Tg ± mice during this time period (*Figure 6a–c*). Conversely, we observed a significant reduction in the number of vacuoles and disrupted photoreceptors in the retina of Tg ± mice, indicating that *Fxn* restoration rescued photoreceptor degeneration (*Figure 6d,e*). These findings establish the principle of cellular dysfunction reversibility in FRDAkd mouse model due to *Fxn* restoration and, therefore, raise the possibility that some neurological and cardiac defects seen in this model and FRDA patients may not be permanent.

In line with remarkable recovery of several behavioral, physiological and pathological defects in FRDAkd mice, we also observed that the genome-wide molecular biomarker represented by gene expression changes accompanying *Fxn* knockdown could be completely rescued after *Fxn* restoration (*Figure 7*). By rescuing the FXN protein levels back to the near basal level, we were able to reverse the molecular changes completely. After 8 weeks of dox removal following an initial 12 weeks of dox treatment, we examined the number of differentially expressed genes (at FDR < 5%) in all tissues and observed 100% normalization of gene expression levels in cerebellum and DRGs, and 99.95% of gene expression levels in the heart of FRDAkd mice (*Figure 7a,b*). These results, which included several pathways (*Figure 7a,c,d*) that are significantly affected due to *Fxn* depletion and that show complete reversal (*Figure 7a,b*) due to *Fxn* restoration may provide a gene-expression signature for evaluation of various therapeutic paradigm.

In summary, all six major behavioral deficits in FRDAkd mice were reversed (*Figure 2*). In the cardiac system, the long QT interval phenotype along with various pathological manifestations, including mitochondrial defects were reversed. In the nervous system, we observed improvement in DRGs and photoreceptor neurons, and complete reversal of the molecular changes in all three tissues suggesting near basal *Fxn* levels are sufficient to alleviate behavioral symptoms in the preclinical FRDAkd model. The rapidity of *Fxn* expression due to dox removal and its robust correction of various parameters, even when restored after the onset of motor dysfunction, makes this FRDAkd mouse model an appealing potential preclinical tool for testing various therapeutics for FRDA.

## Discussion

Here we report development and extensive characterization of a novel, reversible mouse model of FRDA based on knock-down of frataxin by RNA interference (*Seibler et al., 2007*), the FRDAkd mouse. Treated transgenic mice developed abnormal mitochondria, exhibit cardiac and nervous system impairments along with physiological, pathological and molecular deficits. These abnormalities likely contributed to the behavioral phenotype of FRDAkd mice, parallel to what is observed in FRDA patients (*Koeppen and Mazurkiewicz, 2013*; *Koeppen, 2011*). Importantly, restoration of *Fxn* expression, even after the development of severe symptoms and pathological defects, resulted in a drastic amelioration of the clinical phenotype, both in the cardiac and nervous systems, including motor activity. Pathology was significantly improved in the heart, DRGs and in the retina, but only mild improvement was observed in spinal cord, suggesting *Fxn* restoration for the duration of 8 weeks is not sufficient for reversal of neuronal defects in the spinal cord after *Fxn* knockdown. Interestingly, transcriptomic changes due to *Fxn* knockdown in three different tissues differed (*Coppola et al., 2009*), indicating variable response to frataxin deficiency, which could partly explain cell and tissue selectivity in FRDA (*Lynch et al., 2012*). The onset and progression of the disease correlates with the concentration of doxycycline (shRNA induction), and the phenotype returns to baseline after its withdrawal. Since off-target effects due to the shRNA induction are of concern in any RNAi based experiments, we examined the off-target effect by determining the shRNA specificity and measuring changes in gene expression pattern during the induction of shRNA expression targeting the *Fxn* gene. *Fxn* was the earliest and most consistent gene showing significant down-regulation and we did not observe any changes in the RNA levels of other potential targets at any stage in our

analyses, indicative of the absence of significant off-target effects. The onset of transgene expression to achieve *Fxn* knockdown and robust recovery of symptoms due to restoration of *Fxn* levels in a mouse model of FRDA, even when reversed after the onset of the disease, makes this model an appealing potential preclinical tool for developing FRDA therapeutics. This approach will also enable new insights into FRDA gene function and molecular disease mechanisms.

Several models of FRDA have been developed and each have advantages and disadvantages (*Perdomini et al., 2013*). This new FRDAkd model exhibits several unique features that provide advantages for the study of FRDA pathophysiology relative to other existing models (*Miranda et al., 2002*; *Al-Mahdawi et al., 2004*; *Al-Mahdawi et al., 2006*; *Puccio et al., 2001*; *Simon et al., 2004*; *Perdomini et al., 2013*). First, induction of frataxin knockdown permits circumventing potential confounding developmental effects (*Cossée et al., 2000*) and has the flexibility to enhance the disease onset and progression very rapidly by increasing the doxycycline dose. Moreover, the temporal control of *Fxn* knockdown can provide further insights into the sequence of tissue vulnerability during the disease progression (*Lynch et al., 2012*). Second, reversibility of *Fxn* knockdown provides a unique model to mimic the effect of an ideal therapeutic intervention. Along this line, it has so far not been established to what extent FXN restoration after the onset of clinical motor symptoms is successful to prevent occurrence and/or progression of FRDA. Therefore, this model will be of central importance to gain better insights into disease pathogenesis and to test therapeutic agents. Third, the doxycycline-dependent reduction of *Fxn* expression can be tailored to carefully determine the critical threshold of *Fxn* levels necessary to induce selective cellular dysfunction in the nervous system (DRGs and spinal cord) and to understand the occurrence of tissue specific dysfunction in FRDA. Thereby, these experiments will help to understand the tissue specificity and generate clinically relevant tissue targeted therapeutics for FRDA. Finally, the temporal control of single copy and reversible regulation of shRNA expression against *Fxn* produces reproducible transgene expression from the well-characterized rosa26 locus to generate the first model to exhibit and reverse several symptoms parallel to FRDA patients. Here, we focused on the consequences of frataxin removal in an otherwise healthy adult animal. However, we should emphasize that this FRDAkd model uniquely facilitates future studies exploring prenatal or early post natal knockdown, and a wide variety of time course studies to understand disease pathophysiology and to identify potential imaging, physiological, or behavioral correlates of likely reversible or non-reversible disabilities in patients.

We show that FRDAkd animals are defective in several behaviors and exhibit weight loss, reduced locomotor activity, reduced strength, ataxia and early mortality. All of these defects were significantly improved following *Fxn* restoration, approaching or reaching wild-type levels. Most importantly ataxia and survival are well-established and important clinical endpoints in FRDA (*Tsou et al., 2011*), readouts after *Fxn* restoration clearly improve these parameters and appear to be directly related to the functional status of the FRDAkd mice. We conclude that *Fxn* deficient mice exhibit considerable neurological plasticity even in a nervous system that is fully adult. Hence, utilizing these reversible intermediate behavioral phenotypes as biomarkers will help us determine the disease progression and test various FRDA treatment options in this model.

Hypertrophic cardiomyopathy is a common clinical feature in FRDA and approximately 60% of patients with typical childhood onset FRDA die from cardiac failure (*Tsou et al., 2011*). It is generally believed that cardiac failure is caused by the loss of cardiomyocytes through activation of apoptosis (*Fujita and Ishikawa, 2011*). We observed activation of early apoptosis pathways in heart tissue and severe cardiomyopathy characterized by ventricular wall thickness (*Bennett, 2002*). However, we did not observe TUNEL positive cells in either heart or nervous system. This may reflect that the model is in a early phase of cell death initiation, or rather that apoptotic cells are readily phagocytosed by neighboring cells and are consequently difficult to detect (*Ravichandran, 2011*). We also observed enhanced activation of autophagy in the heart tissue of FRDAkd mice, where autophagic cardiomyocytes are observed at a significantly higher frequency during cardiac failure (*Martinet et al., 2007*). These results suggest that apoptosis and autophagy together might synergistically play a vital role in the development of cardiac defect in FRDA (*Eisenberg-Lerner et al., 2009*).

During *Fxn* knockdown, FRDAkd mice initially exhibited a long QT interval at 12 weeks during electrocardiographic analyses, followed by the absence of P-waves and increased ventricular wall thickness at 24 weeks. Restoration of *Fxn* levels at 12 weeks reversed long QT interval phenotype. However, it will be interesting to examine if the ventricular wall thickness can be restored by a more prolonged rescue time period. Another prominent feature of *Fxn* deficiency mouse and FRDA

patients is iron accumulation and deficiency in activity of the iron-sulfur cluster dependent enzyme, aconitase, in cardiac muscle (*Puccio et al., 2001*; *Rötig et al., 1997*; *Delatycki et al., 1999*; *Michael et al., 2006*). Consistent with these observations, we observed increased iron accumulation and reduced aconitase activity in the cardiac tissue of FRDAkd mice and we demonstrate a marked reversal of both to a statistically significant extent, suggesting *Fxn* restoration is sufficient to overcome and clear the iron accumulation and reverse aconitase activity (*Tan et al., 2001*). Our gene expression data revealed several genes (*Hfe* [*Del-Castillo-Rueda et al., 2012*], *Slc40a1* [*Del-Castillo-Rueda et al., 2012*], *Hmox1* [*Song et al., 2012*], *Tfrc* [*Del-Castillo-Rueda et al., 2012*] and *Gdf15* [*Cui et al., 2014*]) directly involved in hemochromatosis and iron overload to be upregulated in our FRDAkd mice, all of which were rescued to normal levels by frataxin restoration. Similarly, several downregulated genes involved in normal cardiac function (*Cacna2D1*, *Abcc9* and *Hrc*) were rescued by *Fxn* restoration. Together, these data indicate that *Fxn* restoration in symptomatic FRDAkd mice reverses the early development of cardiomyopathy at the molecular, cellular and physiological levels.

Cellular dysfunction due to FXN deficiency is presumed to be the result of a mitochondrial defect, since FXN localizes to mitochondria (*Tan et al., 2001*; *Koutnikova et al., 1997*; *Foury and Cazzalini, 1997*) and deficiencies of mitochondrial enzymes and function have been observed in tissues of FRDA patients (*Rötig et al., 1997*; *Lodi et al., 1999*). It has been hypothesized that FXN deficiency induced mitochondrial dysfunction involves the production of ROS, which subsequently lead to cellular dysfunction (*Tamarit et al., 2016*; *Santos et al., 2010*). Several studies have suggested the existence of ROS markers in FRDA patient samples (*Schulz et al., 2000*; *Tozzi et al., 2002*; *Piemonte et al., 2001*; *Emond et al., 2000*). However, other evidence from several different FRDA model organisms suggests that ROS are not elevated (*Chen et al., 2016a*; *Chen et al., 2016b*; *Seznec et al., 2005*; *Shidara and Hollenbeck, 2010*) and more than ten clinical trials based on antioxidant therapy have shown no or limited benefit in FRDA patients (*Kearney et al., 2012*). In our FRDAkd mouse model, we did not observe evidence of a chronic, sustained increase in ROS as measured by three different assays, which is consistent with recent data suggesting that FXN deficiency likely acts via other pathways (*Chen et al., 2016a*; *Chen et al., 2016b*). Here we show that FRDAkd mice displayed accumulation of damaged mitochondria, and reduced aconitase as a direct consequence of frataxin deficiency in heart, consistent with previous findings in conditional FRDA mouse models (*Puccio et al., 2001*; *Simon et al., 2004*). This is consistent with the model whereby frataxin deficiency inhibits mitochondrial function leading to cellular dysfunction (*González-Cabo and Palau, 2013*). Restoration of FXN resulted in improvement in pathological mitochondrial structure indicating that FXN restoration prevents mitochondrial defects and may thereby enhance cell survival (*Tan et al., 2001*). Our observation of mitochondrial dysfunction recovery as early as eight weeks after FXN restoration is consistent with the highly dynamic nature of mitochondrial function (*Chan, 2012*).

In the nervous system, we observed higher level of condensed mitochondria in DRGs through ultrastructural analyses. Condensed mitochondria are known to have lower respiratory control and ATP production (*Desagher and Martinou, 2000*). On most occasions, these condensed mitochondria in DRG neurons of FRDAkd mice were associated with electron-light, or 'empty' vesicles. Degenerating mitochondria in the nervous system can lose their cristae to become empty vesicles co-occurring with either lipid, pigment, and glycogen accumulation, may be the case here (*Golestaneh et al., 2017*). Nile Red staining in the FRDAkd mice was performed to examine if these empty vesicles are lipid bodies. However, while we observed lipid accumulation in these vesicles in the liver, we did not observe detectable levels of lipid staining in the DRGs, heart and spinal cord samples in either control or transgenic animals. Consistent with a mitochondrial origin for these vesicles, we did observe a reduction in condensed mitochondria and their association with these empty vesicles in rescue animals, suggesting that a substantial fraction of dysfunctioning frataxin-deficient mitochondria containing neurons are still viable after the onset of disease and that their dysfunction can be reversed. Additional experiments will be needed to identify the origin and composition of these empty vesicles in this model.

In the spinal cord, we observed reduction of axonal size and myelin sheath thickness in FRDAkd animals, however after eight weeks of rescue period by FXN restoration, we observed limited improvement, suggesting more time may be necessary for improved nervous system recovery. Conversely, disruption of photoreceptor neurons and degenerating RPE cells in the retina displayed

robust recovery, indicating an overall morphological improvement in retinal neurons upon FXN restoration to normal levels. These findings extend previous studies showing that many defects in FRDA cells in vitro and cardiac function in vivo may be reversible following reintroduction of FXN (*Vyas et al., 2012*) by pharmacological interventions targeting aberrant signaling processes (*Rufini et al., 2015*) or by reintroduction of FXN by gene therapy (*Perdomini et al., 2014*).

Improved understanding of the mechanism by which FXN deficiency leads to the various phenotypes observed in FRDA is a major goal of current research in this disorder. In this regard, gene expression analyses identified several pathways altered, including the PPAR signaling pathway, the insulin signaling pathway, fatty acid metabolism, cell cycle, protein modification, lipid metabolism, and carbohydrate biosynthesis, all of which have been previously associated with altered function in FRDA patients (*Coppola et al., 2009*; *Haugen et al., 2010*; *Coppola et al., 2006*). Although we did observe iron deposition associated with *Fxn* knockdown, we did not observe, strong evidence of *Pdk1/Mef2* activation, as previously observed in the fly and a constitutive mouse model (*Chen et al., 2016a*; *Chen et al., 2016b*). This observation may be due to differences in the timing of knockdown. The role of this specific pathway in mediating neurodegeneration needs further exploration following embryonic or early postnatal knockdown in our model and in others. We also observed several immune components, namely, complement and chemokine cascade genes being upregulated, may be implicated as a protective mechanism after *Fxn* knockdown, or in a causal role through chronic activation of the inflammatory response (*Shen et al., 2016*; *Lu et al., 2009*). It will be interesting to assess the causal roles of these genes and pathways and as potent candidate biomarkers for FRDA at several stages during the disease progression.

Determining definitively whether the global expression changes observed in this study are primary or secondary to *Fxn* knockdown will require further investigation by examining more dense time points immediately after *Fxn* knockdown and rescue, but our data clearly support the utility of induced models of disease, whereby the consequences of gene depletion can be more cleanly controlled to examine the molecular changes (*Seibler et al., 2007*). Together with post-induction rescue, this study highlights potential biomarkers and pathways of FRDA progression. For FRDA clinical trials, it will be important to assess whether these expression changes are translatable to the human disease and extend to other tissues that are more easily accessible than CNS or cardiac tissue. Together, these observations suggest that this work also provides a functional genomics foundation for understanding FRDA disease mechanism, progression and recovery.

In conclusion, we characterize a new, inducible model of *Fxn* deficiency and provide multiple lines of evidence that *Fxn* knockdown in adult mice leads to clinical – pathological features parallel to those observed in FRDA patients. By restoring FXN levels we show reversal of several symptoms even after significant motor and cardiac abnormalities, demonstrating that substantial clinical symptoms may result from reversible cellular dysfunction, rather than cell death. These data also highlight the utility of this model for testing potential therapeutics, such as gene therapy (*Perdomini et al., 2014*), protein replacement therapy (*Vyas et al., 2012*), enhancement of mitochondrial function (*Strawser et al., 2014*) and small molecules (*Sandi et al., 2014*; *Kearney et al., 2016*). In fact, our findings suggest that such approaches may not only enhance FXN expression and rescue downstream molecular changes, but may too alleviate pathological and behavioral deficits associated with FRDA, depending on the stage of the disorder.

## Materials and methods

**Key resources table**

| Reagent type (species) or resource | Designation | Source or reference | Identifiers | Additional information |
|---|---|---|---|---|
| strain, strain background (*M. musculus*) | FRDAkd mice (in C57BL/6 background) | this paper | | Frataxin knockdown mice |
| strain, strain background (*M. musculus*) | C57BL/6J | The Jackson Lab | RRID:IMSR_JAX:000664 | |
| cell line (*M. musculus*) | Neuro-2a cell lines (ATCC CCL-131) | ATCC | ATCC CCL-131; RRID:CVCL_0470 | |

*Continued on next page*

*Continued*

| Reagent type (species) or resource | Designation | Source or reference | Identifiers | Additional information |
|---|---|---|---|---|
| cell line (*M. musculus*) | LW-4 (129SvEv) embryonic stem cells | UCLA Transgenic Core facility | | ES cell generated at UCLA from 129SvEv mice |
| chemical compound, drug | Doxycycline (Dox) | Sigma | D9891-100G | Doxycycline hyclate |
| commercial assay or kit | Bradford assay | Bio-Rad | 500–0006 | Bio-Rad Protein Assay Dye Reagent Concentrate |
| commercial assay or kit | Aconitase assay kit | Cayman Chemical | 705502 | |
| commercial assay or kit | Citrate synthase assay kit | Sigma | CS0720 | |
| commercial assay or kit | RNA extraction kit | Qiagen | 217004 | miRNeasy mini kit |
| commercial assay or kit | RNA reverse transcription kit | ThermoFisher Scientific | 18091050 | SuperScript IV First-Strand Synthesis System |
| commercial assay or kit | TUNEL assay | Roche | 11767291910 | in Situ Cell Death Detection Kit, Fluorescein |
| commercial assay or kit | OxyBlot assay | Millipore | S7150 | OxyBlot Protein Oxidation Detection Kit |
| antibody | anti-ferroportin-1 | LifeSpan BioSciences | LS-B1836 RRID:AB_2254938 | Rabbit, 1:200 |
| antibody | anti-ferritin | Abcam | ab69090 RRID:AB_1523609 | Rabbit, 1:500 |
| antibody | anti-LC3 | Abgent | AM1800a RRID:AB_2137696 | Mouse, 1:200 |
| antibody | anti-NeuN | EMD Millipore | ABN91 RRID:AB_11205760 | Chicken, 1:500 |
| antibody | anti-NeuN | EMD Millipore | ABN78 RRID:AB_10807945 | Rabbit, 1:500 |
| antibody | Alexa Fluor 488 | ThermoFisher Scientific | A-11008 RRID:AB_143165 | Goat anti-rabbit IgG, 1:500 |
| antibody | Alexa Fluor 594 | ThermoFisher Scientific | A-11039 RRID:AB_142924 | Goat anti-chicken IgG, 1:500 |
| antibody | Alexa Fluor 488 | ThermoFisher Scientific | A-11029 RRID:AB_138404 | Goat anti-mouse IgG, 1:500) |
| antibody | anti-Frataxin | Santa Cruz biotechnology | sc-25820 RRID:AB_2110677 | Rabbit, 1:200 |
| antibody | anti-Akt | Cell Signaling Technology | 4691 RRID:AB_915783 | Rabbit, 1:500 |
| antibody | anti-Caspase 8 | Cell Signaling Technology | 8592S RRID:AB_10891784 | Rabbit, 1:200 |
| antibody | Anti-b-Actin | Sigma-Aldrich | A1978 RRID:AB_476692 | Mouse, 1:1000 |
| antibody | Anti-4-Hydroxynonenal | Abcam | ab46545 RRID:AB_722490 | Rabbit, 1:3000 |
| antibody | Anti-4-Nitrotyrosine | Abcam | ab61392 RRID:AB_942087 | Mouse, 1:3000 |
| antibody | Anti-phospho-PDK1 | Cell Signaling Technology | 3438S RRID:AB_2161134 | Rabbit, 1:1000 |
| antibody | Goat-anti-rabbit HRP-conjugated secondary antibody | ThermoFisher Scientific | 31460 RRID:AB_228341 | 1:5000 |
| antibody | Goat-anti-mouse HRP-conjugated secondary antibody | ThermoFisher Scientific | 32230 RRID:AB_1965958 | 1:5000 |
| software, algorithm | ImageJ | https://imagej.nih.gov/ij/ | RRID:SCR_003070 | |

*Continued*

| Reagent type (species) or resource | Designation | Source or reference | Identifiers | Additional information |
|---|---|---|---|---|
| software, algorithm | BLASTN | https://blast.ncbi.nlm.nih.go | RRID:SCR_001598 | |
| software, algorithm | TopScan | http://cleversysinc.com/?csi_products=topscan-lite | RRID:SCR_014494 | |
| software, algorithm | ImageScope | http://www.aperio.com/ | RRID:SCR_014311 | |
| software, algorithm | CellProfiler Analyst | http://cellprofiler.org | RRID:SCR_010649 | |
| software, algorithm | LIMMA | http://bioinf.wehi.edu.au/limma/ | RRID:SCR_010943 | |
| software, algorithm | WGCNA | http://www.genetics.ucla.edu/labs/horvath/CoexpressionNetwork/ | RRID:SCR_003302 | Weighted Gene Co-expression Network Analysis |
| software, algorithm | DAVID tools | http://david.abcc.ncifcrf.gov/ | RRID:SCR_003033 | Database for Annotation Visualization and Integrated Discovery |
| software, algorithm | PubMatrix | http://pubmatrix.grc.nia.nih.gov/ | RRID:SCR_008236 | |
| software, algorithm | AccessPoint | http://freelandsystems.com/accesspoint-suite/accesspoint/ | RRID:SCR_015792 | |
| software, algorithm | NOTOCORD-hem | http://www.notocord.com/software/notocord_hem | RRID:SCR_015793 | |

## In vitro frataxin knockdown

Six different shRNA sequences against the frataxin mRNA (*Figure 1—figure supplement 1*) were cloned into the exchange vector (proprietary material obtained from TaconicArtemis GmbH) as described in detail previously (*Seibler et al., 2007*). To increase confidence in our selected shRNA sequence, we performed independent experiments utilizing six different shRNA sequence targeting *Fxn*. We measured *Fxn* mRNA and protein levels in vitro after over-expression of these shRNA sequences. N2A cells (ATCC CCL-131; RRID:CVCL_0470) were transduced with exchange vectors containing the shRNA against frataxin using Fugene 6 (Promega, Madison, WI). After 24 hr, cells were replated in media containing neomycin and were validated for efficient frataxin knockdown by RT-PCR and Western blotting. All six shRNA sequence displayed significant *Fxn* knockdown, suggesting the possibility of limited off-target effects, since multiple shRNAs targeting the same gene (*Fxn*) produce comparable gene silencing efficacy.

We selected the most efficient shRNA (GGATGGCGTGCTCACCATTAA) for validation and generating the transgenic animal. To examine potential off-target effects more directly, we determined the shRNA specificity by measuring global changes in gene expression pattern following the induction of shRNA expression targeting *Fxn* in our in vivo model. First, we screened and identified potential putative targets of the shRNA sequence (GGATGGCGTGCTCACCATTAA) across the entire mouse genome utilizing the BLASTN program (RRID:SCR_001598). This identified 20 unique genes as potential targets in the mouse genome, which included *Fxn* gene as top hit, with all 21 base pairs matching as expected (*Figure 1—figure supplement 2*). The remaining potential targets had only 16 to 13 base pair matches. Next, we examined the gene expression levels for each potential target gene present in our microarray data after induction of the shRNA. *Fxn* was the earliest and most consistent gene showing significant down-regulation. We did not observe any changes in the RNA levels of other potential targets at any stage in our gene expression analysis (*Figure 1—figure supplement 2*), strong evidence against significant off-target effects.

Mouse LW-4 (129SvEv) embryonic stem cells were cultured and the rosa26 targeting vector (from TaconicArtemis GmbH) were electroporated according to protocol described before (*Seibler et al., 2005*). The exchange vector containing validated and selected shRNA sequence (GGATGGCGTGCTCACCATTAA) against the *Fxn* mRNA were electroporated to obtain positive ES cells containing shRNA expression cassette integrated into the ROSA26 locus. Correctly targeted embryonic stem cell clones were utilized to generate frataxin knockdown mice (see below).

## Transgenic mouse generation

Transgenic mice were generated at UCLA Transgenic Core facility using proprietary materials obtained from TaconicArtemis GmbH (Köln, Germany). In brief, mouse LW-4 (129SvEv) embryonic stem cells with the recombinase-mediated cassette exchange (RMCE) acceptor site on chromosome 6 were used for targeting insertion of distinct Tet-On frataxin shRNA expression cassettes into the ROSA26 locus (*Seibler et al., 2007*) as depicted in *Figure 1*. Correctly targeted embryonic stem cell clones were identified by Southern DNA blot analysis and tested for frataxin mRNA knockdown at the embryonic stem cell stage (not shown). One embryonic stem cell clone that gave acceptable mRNA knockdown was microinjected into C57BL/6J blastocysts from which chimeric mice were derived. These frataxin knockdown mice (FRDAkd) were backcrossed six generations into the C57BL/6 mouse background (RRID:IMSR_JAX:000664).

## Genotyping

Mouse tail biopsies were collected and DNA was extracted in boiling buffer (25 mM sodium hydroxide, 0.2 mM EDTA) at 98°C for 60 min. Extracted DNA was neutralized in Tris/HCl buffer (pH5.5) and PCR was performed under the following conditions with BioMix Red (Bioline). Four primers were used in the reaction: 5'-CCATGGAATTCGAACGCTGACGTC-3', 5'-TATGGGCTATGAACTAATGACCC-3', to amplify shRNA; 5'-GAGACTCTGGCTACTCATCC-3', 5'- CCTTCAGCAAGAGCTGGGGAC-3', as genomic control. The cycling conditions for PCR amplification were: 95°C for 5 min; 95°C for 30 s, 60°C for 30 s, 72°C for 1 min, (35 cycles); 72°C for 10 min. PCR products were analyzed by gel electrophoresis using 1.5% agarose and visualized by Biospectrum imaging system (UVP).

## Animal and study design

Experiments were approved by the Animal Research Committee (ARC) of University of California, Los Angeles and were in accordance with ARC regulation. Extensive neurological and neuropsychological tests (body weight, poorly groomed fur, bald patches in the coat, absence of whiskers, wild-running, excessive grooming, freezing, hunched body posture when walking, response to object [cotton-tip swab test], visual cliff behavior analysis) were performed to ensure all animals included in the studies were healthy. Age and sex were matched between wild type (Wt) and transgenic (Tg) groups to eliminate study bias. The average age of the animals at the start of experiments was 3–4 months. Three different study cohorts were implemented: behavior, pathology and gene expression. Animals were randomly assigned to different experimental groups with in these three different study cohorts before the start of experiments. Due to high mortality rate of FRDAkd animals, we doubled the Tg + animal number (a sample size of 30 animals per Tg + treatment group for behavioral analyses (60 total)) in order to have sufficient power for statistical analyses. All the group size for our experiments were determined by statistical power analysis. The values utilized were: the power = 0.9, alpha = 0.05, Coefficient of determination = 0.5, effect size = 0.70. Effect size and power calculations were based on our pilot experiments.

## In vivo frataxin knockdown

Animals were divided into the following groups: Wt treated with doxycycline (Dox) (Wt +), Wt without Dox (Wt -), Tg treated with Dox (Tg +), Tg without Dox (Tg -), Tg Dox rescue (Tg ±). First, we examined the Tg + mice for frataxin knockdown utilizing higher dose of dox (4 and 6 mg/ml), we observed mortality as early as two weeks and a 100% mortality rate by 5 to 6 weeks (not shown). To avoid early mortality and to have slow and steady state of disease progression we followed 2 mg/ml in drinking water coupled with intraperitoneal injection of dox (5 or 10 mg/kg) twice per week. Doxycycline (2 mg/mL) was added to the drinking water of all treatment animals which was changed weekly. In addition, animals were injected intraperitoneally (IP) with Dox (5 mg/kg body weight) twice a week for 10 weeks followed by 10 mg Dox/kg body weight twice a week for 2 weeks. Animals in Tg Dox rescue (Tg ±) group were given untreated water and not injected with Dox after week 12. All animals were weighed weekly.

**Behavior cohort**

## Animal information

A total of 108 animals (Wt n = 32, Tg n = 76) were included in this study with equal numbers of male and female animals. For all tests, investigators were blinded to genotype and treatment. For all behavioral tests, the variance between all the groups for that specific behavioral test were observed to be initially not statistically significant.

## Accelerating rotarod

To measure motor function, rotarod analysis was performed weekly at the start of Dox treatment using an accelerating rotorod (ROTOMEX-5, Columbus Instruments, Columbus, OH). Mice were assessed for 36 weeks. Briefly, after habituation, a mouse was placed on the rotarod rotating at 5 rpm for one min and then the rotorod was accelerated at 0.09 rpm/sec$^2$. The latency to fall from a rotating rod after acceleration was recorded. Each mouse was subjected to three test trials within the same day with a 15 min inter-trial interval. The average latency normalized by the mouse body weight in the test week was used for data analysis.

## Grip strength test

The grip strength was measured at week 0, 12 and 24 weeks using a digital force gauge (Chatillon Force Measurement Systems, AMETEK TCI Division, Largo, FL). Briefly, the mouse was allowed to grasp the steel wired grid attached to the force gauge with only forepaws. The mouse was then pulled back from the gauge. The force applied to the grip immediately before release of the grip is recorded as the 'peak tension' and is a measurement of forepaw strength. The same measurement was repeated with the mouse grasping the grid with all paws for whole grip strength. Each mouse was subjected to three forepaw and whole strength measurements. The hindpaw grip strength was calculated as the average whole grip strength minus average forepaw strength. The value of hind-paw grip strength normalized by the mouse body weight in the test week was used for data analysis (*Crawley, 2007*).

## Gait analyses

Gait analysis was performed at week 0, 12 and 24 weeks by allowing the animals to walk through a 50-cm-long, 10-cm-wide runway that was lined with blank index cards. After a period of habituation (walking through the runway three times), hind and fore paws was coated with nontoxic red and purple paint respectively and the mouse was allowed to walk through the runway again. Footprints were captured on the index cards. The index cards were scanned and the images were measured in image J (RRID:SCR_003070) for calculating the stride length and other parameters as per (*Carter et al., 2001*).

## Open field analyses

Open field activity monitoring system was used to assess the level of spontaneous locomotion and behavioral activity. Open field testing was performed at week 0, 12 and 24 weeks in an open field Plexiglas chamber with a video monitoring system that records any movement of mouse within the chamber. Mice were placed in the chambers for 5 min to acclimate. The recording system was turned on without disturbing the animals in the testing chambers. TopScan software (RRID:SCR_014494) connected to the chamber was used to track the behavior of the animal during the testing. Each mouse placed in different quadrants of open field chambers was recorded for 20 min. After the end of the recording period the mouse was returned to their respective home cages. Locomotor activity of the mice was analyzed by TopScan (CleverSys; RRID:SCR_014494) software.

**Pathology cohort**

## Animal information

A total of 57 animals (Wt n = 21 (10 females, 11 males), Tg n = 36 (20 females, 16 males)) were euthanized in the study. Animals were deeply anesthetized with sodium pentobarbital (40 mg/kg body weight) and perfused intracardially with 20 mL PBS (5 mL/min) followed by 20 mL freshly made 4% paraformaldehyde (5 mL/min). Tissue from liver, lung, spleen, pancreas, kidney, heart, eye

(retina), brain, muscle, spinal cord, dorsal root ganglion (DRG) and sciatic nerve was dissected and collected immediately after perfusion. Collected tissue was immersed in 4% paraformaldehyde overnight at 4°C and then transferred and kept in 40% sucrose at 4°C until embedded with O.C.T. Tissue cryosections (5 μM) were collected with a cryostat for staining.

## Staining

H&E staining was performed in the Translational Pathology Core Laboratory at UCLA using methods described briefly below. After equilibration, the tissue section was stained with Harris Modified Hematoxylin for 10 min, then in Eosin Y for 10 min. Gomori's iron staining and Masson's trichrome staining were performed in Histopathology lab in UCLA, the procedure is described briefly as below. Gomori's iron staining: the tissue section was immersed for 20 min in equal parts of 20% hydrochloric acid and 10% potassium ferrocyanide, washed thoroughly in distilled water. After counterstaining with Nuclear Fast Red (0.1% Kernechtrot in 5% aluminum sulfate), the slides were dehydrated through gradual ethyl alcohol solutions and mounted for imagining. Masson's trichrome staining: the tissue section was fixed in Bouin's fixative (75 mL picric acid saturated aqueous solution, 25 mL of 40% formaldehyde, 5 mL glacial acetic acid) for 60 min at 60°C, stained with Weigert iron hematoxylin (0.5% hematoxylin, 47.5% alcohol, 0.58% ferric chloride, 0.5% hydrochloric acid) for 10 min, Biebrich scarlet-acid fuchsin (0.9% Biebrich scarlet, 0.1% acid fuchsin, 1% glacial acetic acid) for 10 min, immersed in phosphomolybdic-phosphotungstic solution (2.5% phosphomolybdic acid, 2.5% phosphotungstic acid) for 7 min, stained with aniline blue solution (2.5% aniline blue, 2% acetic acid) for 7 min, dehydrate and mounted for imagining. Immunofluorescence staining was performed as followed: cryosections were equilibrated with TBS (20 mM Tris pH7.5, 150 mM NaCl) and permeabilized in TBST (TBS with 0.2% TritonX-100). Tissue sections were incubated with primary antibody in TBST with 4% normal goat serum (Jackson Immuno Research) (4°C, overnight), followed by incubation with secondary antibody in TBST with 4% normal goat serum (room temperature, 2 hr) the next day. The tissue section was mounted with ProLong Gold with DAPI (ThermoFisher Scientific) and stored in the dark. The following antibodies were used: anti-ferroportin-1 (LifeSpan BioSciences, LS-B1836, rabbit, 1:200; RRID:AB_2254938), anti-ferritin (Abcam, ab69090, rabbit, 1:500; RRID:AB_1523609), anti-LC3 (Abgent, AM1800a, mouse, 1:200; RRID:AB_2137696), anti-NeuN (EMD Millipore, ABN91, chicken, 1:500; RRID:AB_11205760), anti-NeuN (EMD Millipore, ABN78, rabbit, 1:500; RRID:AB_10807945). Secondary antibodies conjugated with Alexa Fluor (ThermoFisher Scientific, 1:500) were used as indicated: Alexa Fluor 488 (goat anti-Rabbit IgG, A-11008; RRID:AB_143165), Alexa Fluor 594 (goat anti-chicken IgG, A-11039; RRID:AB_142924), and Alexa Fluor 488 (goat anti-mouse IgG, A-11029; RRID:AB_138404). DNA strand breaks were determined by TUNEL assay (Roche). Briefly, sections were equilibrated with PBS, permeabilized in 0.1% Triton X-100 in 0.1% sodium citrate (on ice, 2 min) and incubated with TUNEL reaction mixture in a humidified atmosphere (37°C, 60 min). Sections were mounted with ProLong Gold with DAPI and stored in the dark.

## Imaging and quantification

Slides stained for hematoxylin-eosin, iron, and trichrome were scanned (20 x magnification) by Aperio ScanScope with Aperio ImageScope Software (Leica; RRID:SCR_014311). Immunofluorescence staining and TUNEL staining were imaged with LSM780 confocal microscope system (Zeiss). Images were quantified by CellProfiler (RRID:SCR_010649). For Purkinje cell quantification, the H and E images were scanned, visualized and exported by ImageScope (Leica; RRID:SCR_014311) system. Exported images were utilized to count the Purkinje cells manually using NIH ImageJ software (RRID:SCR_003070). Counting was conducted by investigators who were blinded to genotype and treatment.

## Electron microscopy analyses

A total of 29 animals (Wt n = 9 (four females, five males), Tg n = 20 (9 females, 11 males)) were used in the study. Mice were perfused transcardially with 2.5% glutaraldehyde, 2% paraformaldehyde in 0.1M phosphate buffer, 0.9% sodium chloride (PBS). Pieces of heart, lumbar spinal cord, dorsal root ganglia, muscle, and eye were dissected, postfixed for 2 hr at room temperature in the same fixative, and stored at 4°C until processing. Tissues were washed with PBS, postfixed in 1% OsO4 in PBS for 1 hr, dehydrated in a graded series of ethanol, treated with propylene oxide and infiltrated with

Eponate 12 (Ted Pella) overnight. Tissues were embedded in fresh Eponate, and polymerized at 60°C for 48 hr. Approximately 60–70 nm thick sections were cut on a RMC Powertome ultramicrotome and picked up on formvar coated copper grids. The sections were stained with uranyl acetate and Reynolds lead citrate and examined on a JEOL 100CX electron microscope at 60 kV. Images were collected on type 4489 EM film and the negatives scanned to create digital files. These high quality digital images were utilized to quantify the number of condensed mitochondria. Condensed mitochondria, vesicles with condensed mitochondria, and vesicles alone were manually counted with NIH ImageJ software (RRID:SCR_003070). Animal genotype and treatment information was blinded to the person who conducted the evaluation.

## Gene expression cohort

### Sample collection

A total of 80 animals (Wt n = 24 (12 females and 12 males), Tg n = 56 (31 females and 25 males)) were sacrificed and tissue was collected in this study. Animals were sacrificed at week 0, 3, 8, 12, 16, 20, and plus 4 and 8 weeks post dox treatment (rescue). Mice were sacrificed by cervical dislocation and tissue from liver, lung, spleen, pancreas, kidney, heart, eye (retina), brain, muscle, spinal cord, dorsal root ganglion (DRG) and sciatic nerve was dissected and rinsed in cold PBS quickly (3X) to remove blood. Tissue samples were transferred immediately into 2 mL RNase-free tubes and immersed into liquid nitrogen. The collected tissue was stored at −80°C immediately.

### RNA extraction

Heart, cerebellum and DRG neuron samples from week 0, 3, 12, 16, 20 and 4, 8 weeks post dox treatment (rescue), each with four biological replicates, were used for expression profiling. Samples were randomized prior to RNA extraction to eliminate extraction batch effect. Total RNA was extracted using the miRNeasy mini kit (Qiagen) according to manufacturer's protocol and including an on-column DNase digest (RNase free DNAse set; Qiagen). RNA samples were immediately aliquoted and stored at −80°C. RNA concentration and integrity were later determined using a Nanodrop Spectrophotometer (ThermoFisher Scientific) and TapeStation 2200 (Agilent Technologies), respectively.

### Transcriptome profiling by microarray

One hundred nanograms of RNA from heart and cerebellum tissue was amplified using the Illumina TotalPrep-96 RNA Amplification kit (ThermoFisher Scientific) and profiled by Illumina mouse Ref 8 v2.0 expression array chips. For DRG samples 16.5 ng of RNA was amplified using the Ovation PicoSL WTA System V2 kit (NuGEN). Only RNA with RIN greater than 7.0 was included for the study. A total of 64 RNA samples for each tissue (n = 192 arrays) were included and samples were randomized before RNA amplification to eliminate microarray chip batch effect. Raw data was log transformed and checked for outliers. Inter-array Pearson correlation and clustering based on variance were used as quality-control measures. Quantile normalization was used and contrast analysis of differential expression was performed by using the LIMMA package (*Smyth, 2005*; RRID:SCR_010943). Briefly, a linear model was fitted across the dataset, contrasts of interest were extracted, and differentially expressed genes for each contrast were selected using an empirical Bayes test statistic (*Smyth, 2005*).

### Construction of co-expression networks

A weighted signed gene co-expression network was constructed for each tissue dataset to identify groups of genes (modules) associated with temporal pattern of expression changes due to frataxin knockdown and rescue following a previously described algorithm (*Zhang and Horvath, 2005*; *Oldham et al., 2006*; RRID:SCR_003302). Briefly, we first computed the Pearson correlation between each pair of selected genes yielding a similarity (correlation) matrix. Next, the adjacency matrix was calculated by raising the absolute values of the correlation matrix to a power (β) as described previously (*Zhang and Horvath, 2005*). The parameter β was chosen by using the scale-free topology criterion (*Zhang and Horvath, 2005*), such that the resulting network connectivity distribution best approximated scale-free topology. The adjacency matrix was then used to define a measure of node dissimilarity, based on the topological overlap matrix, a biologically meaningful

measure of node similarity (*Zhang and Horvath, 2005*). Next, the probe sets were hierarchically clustered using the distance measure and modules were determined by choosing a height cutoff for the resulting dendrogram by using a dynamic tree-cutting algorithm (*Zhang and Horvath, 2005*).

## Consensus module analyses

Consensus modules are defined as sets of highly connected nodes that can be found in multiple networks generated from different datasets (tissues) (*Chandran et al., 2016*). Consensus modules were identified using a suitable consensus dissimilarity that were used as input to a clustering procedure, analogous to the procedure for identifying modules in individual sets as described elsewhere (*Langfelder and Horvath, 2007*). Utilizing consensus network analysis, we identified modules shared across different tissue data sets after frataxin knockdown and calculated the first principal component of gene expression in each module (module eigengene). Next, we correlated the module eigengenes with time after frataxin knockdown to select modules for functional validation.

## Gene ontology, pathway and PubMed analyses

Gene ontology and pathway enrichment analysis was performed using the DAVID platform (DAVID, https://david.ncifcrf.gov/ (*Huang et al., 2008*); RRID:SCR_003033). A list of differentially regulated transcripts for a given modules were utilized for enrichment analyses. All included terms exhibited significant Benjamini corrected P-values for enrichment and generally contained greater than five members per category. We used PubMatrix (*Becker et al., 2003*); RRID:SCR_008236) to examine each differentially expressed gene's association with the observed phenotypes of FRDAkd mice in the published literature by testing association with the key-words: ataxia, cardiac fibrosis, early mortality, enlarged mitochondria, excess iron overload, motor deficits, muscular strength, myelin sheath, neuronal degeneration, sarcomeres, ventricular wall thickness, and weight loss in the PubMed database for every gene.

## Data availability

Datasets generated and analyzed in this study are available at Gene Expression Omnibus. Accession number: GSE98790. R codes utilized for data analyses are available in the following link: https://github.com/dhglab/FxnMice

# Quantitative real-time PCR

RT-PCR was utilized to measure the mRNA expression levels of frataxin in order to identify and validate potent shRNA sequence against frataxin gene. The procedure is briefly described below: 1.5 μg total RNA, together with 1.5 μL random primers (ThermoFisher Scientific, catalog# 48190–011), 1.5 μL 10 mM dNTP (ThermoFisher Scientific, catalog# 58875) and RNase-free water up to 19.5 μL, was incubated at 65°C for 5 min, then on ice for 2 min; 6 μL first strand buffer, 1.5 μL 0.1 M DTT, 1.5 μL RNaseOUT (ThermoFisher Scientific, catalog# 100000840) and 1.5 μL SuperScript III (ThermoFisher Scientific, catalog# 56575) were added to the mixture and incubated for 5 min at 25°C, followed by 60 min at 50°C, and 15 min at 70°C. The resulted cDNA was diluted 1:10 and 3 μL was mixed with 5 μL SensiFast SYBR No-ROX reagent (Bioline, catalog# BIO-98020), 0.6 μL primer set (forward and reverse, 10 mM) and 1.4 μL nuclease free water for real-time PCR. Each reaction was repeated three times on LightCycler 480 II (Roche, catalog# 05015243001). To check *Mef2* activity, the regulation of its target downstream genes were determined by qPCR on cerebellum and heart RNA samples at week 20, utilizing previously validated primers (*Chen et al., 2016b*) and the following thermocycling conditions: 95°C 5 min; 95°C 10 s, 60°C 10 s, 72°C 10 s, repeat 45 cycles; 95°C 5 s, 65°C 1 min, 97°C 5 min.

# Western blotting

Prior to samples being analyzed by Western blotting, protein concentration was estimated by Bradford assay (Bio-Rad). Proteins were separated by SDS-PAGE and transferred onto PVDF membranes. The membranes were incubated with primary and secondary antibodies for protein detection. The target bands were detected with antibodies: anti-Frataxin (Santa Cruz Biotechnology, sc-25820, rabbit, 1:200; RRID:AB_2110677), anti-Akt (Cell Signaling Technology, #4691, rabbit, 1:500; RRID:AB_915783), anti-β-Actin (Sigma, #A1978, mouse, 1:1000; RRID:AB_476692), anti-Caspase 8 (rabbit

monoclonal, Cell Signaling Technology, #8592S, rabbit, 1:200; RRID:AB_10891784), anti-LC3 (Abgent, # AM1800a, mouse, 1:200; RRID:AB_2137696) and goat-anti-rabbit or goat-anti-mouse HRP-conjugated secondary antibodies (ThermoFisher Scientific, #31460; RRID:AB_228341, and #32230; RRID:AB_1965958, respectively,1:5000).

To determinate the ROS level, three different ROS related protein modifications were measured in three different tissue types (brain, muscle and liver) at two different time points (wk12 an wk20). Briefly, anti-4-Hydroxynonenal (Abcam, ab46545, rabbit, 1:3000) antibody was used to detect aldehydic products of lipid peroxidation; anti-3-Nitrotyrosine (Abcam, ab61392, mouse, 1:3000) antibody was used to detect protein tyrosine nitration, an important component of nitric oxide signaling; and OxyBlot Protein Oxidation Detection Kit (Millipore, Cat#S7150) was used to detect the carbonyl groups of protein side, a product of protein oxidative modification. To determinate the phosphorylation of PDK1, anti-pPDK1 (Cell Signaling Technology, #3438, rabbit, 1:1000), and goat-anti-rabbit HRP-conjugated secondary antibody (ThermoFisher Scientifc, #31460, 1:5000) were used. Image J was used for band quantification.

## Enzyme activity assay

Proteins from heart tissue for all genotypes at week 20 was extracted in Tris buffer (50 mM Tris-HCl pH 7.4). Around 80 mg of tissue was homogenized with 100 µL Tris buffer, and centrifuged at 800 g for 10 min at 4°C. The supernatant was transferred to a clean tube and stored at −80°C freezer immediately for later use. Protein concentration was estimated by Bradford assay. Total of 5 µg of protein was used to determine aconitase activity by aconitase assay kit (Cayman Chemical, catalog# 705502). The absorbance of the reaction mixture at 340 nm was measured every minute for 60 min at 37°C, by Synergy-2 (BioTek) plate reader. The linear phase was used to calculate the enzyme activity. 5 µg of protein was used to measure citrate synthase activity by citrate synthase assay kit (Sigma, catalog# CS0720). The absorbance of the reaction mixture at 412 nm was measured by Synergy-2 plate reader every 31 s for 30 min at room temperature. The linear phase was used to calculate the enzyme activity. Aconitase activity in each sample was normalized to the citrate synthase activity from the same sample for comparison.

## Echocardiography

Echocardiography was performed with a Siemens Acuson Sequoia C256 instrument (Siemens Medical Solutions, Mountain View, California). The mice were sedated with isoflurane vaporized in oxygen (Summit Anesthesia Solutions, Bend Oregon). Left ventricular dimensions (EDD, end diastolic dimension; ESD, end systolic dimension; PWT, posterior wall thickness; VST, ventricular septal thickness) were obtained from 2D guided M-mode and was analyzed using AccessPoint software (Freeland System LLC, Santa Fe, New Mexico; RRID:SCR_015792) during systole and diastole. Left ventricular mass was calculated as described previously (*Tanaka et al., 1996*). Heart rate, aortic ejection time, aortic velocity and mitral inflow E and A wave amplitudes were determined from Doppler flow images. Indices of contractility such as left ventricular fractional shortening (LVFS), velocity of circumferential fiber shortening (VCF) and ejection fraction (EF) was obtained from the images. During the procedure, heart rates were maintained at physiological levels by monitoring their electrocardiograms (ECG).

## Surface ECG

Electrocardiogram (ECG) were obtained in all the mice under Isoflurane anesthesia by inserting two Pt needle electrodes (Grass Technologies, West Warwick, RI) under the skin in the lead II configuration as described previously (*Nakashima et al., 2014*). The mice were studied between 10 and 30 min to elucidate any rhythm alteration. The ECG data were amplified (Grass Technologies) and then digitized with HEM V4.2 software (Notocord Systems, Croissy sur Seine, France; RRID:SCR_015793). Measurements of HR, PR, RR, QRS, QT intervals were obtained for comparison and statistical analysis among all the mice groups.

## Accession codes

Gene Expression Omnibus: Datasets generated and analyzed in this study are available at GEO accession: GSE98790.

## Acknowledgements

We gratefully acknowledge support from the Friedreich's Ataxia Research Alliance to DHG and VC (including a New Investigator Award to VC), the Muscular Dystrophy Association to VC and the Dr. Miriam and Sheldon G Adelson Medical Research Foundation to DHG. We thank Marianne Cilluffo for expert assistance with electron microscopy at the BRI Electron Microscopy Core Facility at UCLA.

## Additional information

### Funding

| Funder | Grant reference number | Author |
| --- | --- | --- |
| Friedreich's Ataxia Research Alliance | FARA New Investigator Award | Vijayendran Chandran Daniel H Geschwind |
| Dr. Miriam and Sheldon G. Adelson Medical Research Foundation | Adelson Program in Neural Repair and Regeneration (APNRR) | Daniel H Geschwind |
| Muscular Dystrophy Association | Research Grant | Vijayendran Chandran |

The funders had no role in study design, data collection and interpretation, or the decision to submit the work for publication.

### Author contributions

Vijayendran Chandran, Conceptualization, Data curation, Formal analysis, Supervision, Funding acquisition, Validation, Investigation, Visualization, Methodology, Writing—original draft, Project administration, Writing—review and editing; Kun Gao, Data curation, Formal analysis, Validation, Investigation, Methodology, Project administration; Vivek Swarup, Formal analysis, Investigation, Visualization, Methodology, Project administration; Revital Versano, Data curation, Formal analysis, Validation, Investigation, Visualization, Methodology, Project administration; Hongmei Dong, Maria C Jordan, Investigation, Methodology, Project administration; Daniel H Geschwind, Conceptualization, Supervision, Funding acquisition, Investigation, Methodology, Writing—original draft, Project administration, Writing—review and editing

### Author ORCIDs

Vijayendran Chandran (iD) https://orcid.org/0000-0002-2469-6263
Daniel H Geschwind (iD) https://orcid.org/0000-0003-2896-3450

### Ethics

Animal experimentation: Animal experimentation: This study was performed in strict accordance with the recommendations in the Guide for the Care and Use of Laboratory Animals of the National Institutes of Health. All of the animals were handled according to approved institutional Animal Research Committee (ARC) protocol 200312841 of University of California, Los Angeles.

### Decision letter and Author response

Decision letter https://doi.org/10.7554/eLife.30054.054
Author response https://doi.org/10.7554/eLife.30054.055

## Additional files

### Supplementary files

• Supplementary file 1. Differentially expressed genes after frataxin knockdown and rescue. Genes with a significant differential expression in frataxin knockdown mice versus control in heart, cerebellum and DRG tissues are provided.
DOI: https://doi.org/10.7554/eLife.30054.034

• Supplementary file 2. Gene ontology analysis of the genes that are differentially regulated in frataxin knockdown mice. DAVID analysis software was used to find the Gene Ontology (GO) terms that were significantly enriched (Benjamini corrected p-value<0.05) in differentially regulated genes from FRDAkd mice when compared to the controls.
DOI: https://doi.org/10.7554/eLife.30054.035

• Supplementary file 3. Gene-module membership association based on WGCNA co-expression networks. Twenty consensus modules shared between the three tissues (heart, cerebellum and DRGs) obtained from FRDAkd mice expression data are denoted along with genes in these modules and kME values (Intramodular connectivity).
DOI: https://doi.org/10.7554/eLife.30054.036

• Supplementary file 4. Gene ontology analysis of frataxin knockdown and rescue associated modules. For categorization and clustering of GO terms, we considered GO terms with Benjamini-corrected P-values less than 0.05. Enriched GO terms are provided for each module in separate tabs.
DOI: https://doi.org/10.7554/eLife.30054.037

• Supplementary file 5. Broad functional categories of combined GO ontology terms associated with frataxin knockdown and rescue. The GO ontology terms (*Supplementary file 4*) which are enriched in the 11 modules were combined into 26 broad functional categories based on GO slim hierarchy are provided.
DOI: https://doi.org/10.7554/eLife.30054.038

• Supplementary file 6. Literature annotation of genes associated with the observed phenotype in FRDAkd mice. Table providing differentially expressed gene's association with observed phenotype in FRDAkd mice based on the published literature by testing association with the key-words: ataxia, cardiac fibrosis, early mortality, enlarged mitochondria, excess iron overload, motor deficits, muscular strength, myelin sheath, neuronal degeneration, sarcomeres, ventricular wall thickness, and weight loss in the PubMed database for every gene. The total number of hits (publications) for each gene are represented.
DOI: https://doi.org/10.7554/eLife.30054.039

• Transparent reporting form
DOI: https://doi.org/10.7554/eLife.30054.040

## Major datasets

The following dataset was generated:

| Author(s) | Year | Dataset title | Dataset URL | Database, license, and accessibility information |
|---|---|---|---|---|
| Chandran V, Gao K, Swarup V, Versano R, Dong H, Jordan MC, Geschwind DH | 2017 | Gene expression changes due to frataxin deficiency and restoration in frataxin knockdown mouse model. | https://www.ncbi.nlm.nih.gov/geo/query/acc.cgi?acc=GSE98790 | Publicly available at the NCBI Gene Expression Omnibus (accession no: GSE98790) |

The following previously published datasets were used:

| Author(s) | Year | Dataset title | Dataset URL | Database, license, and accessibility information |
|---|---|---|---|---|
| Huang ML, Richardson DR | 2011 | Expression data of MCK conditional frataxin knock-out mice | https://www.ncbi.nlm.nih.gov/geo/query/acc.cgi?acc=GSE31208 | Publicly available at the NCBI Gene Expression Omnibus (accession no: GSE31208) |
| Coppola G, Marmolino D, Lu D, Wang Q, Cnop M, Rai M, Acquaviva F, Cocozza S, Pandolfo M, Geschwind DH | 2009 | Functional genomic analysis of frataxin deficiency, Agilent data | https://www.ncbi.nlm.nih.gov/geo/query/acc.cgi?acc=GSE15843 | Publicly available at the NCBI Gene Expression Omnibus (accession no: GSE15843) |

| Coppola G, Marmolino D, Lu D, Wang Q, Cnop M, Rai M, Acquaviva F, Cocozza S, Pandolfo M, Geschwind DH | 2009 | Functional genomic analysis of frataxin deficiency, Illumina data | https://www.ncbi.nlm.nih.gov/geo/query/acc.cgi?acc=GSE15848 | Publicly available at the NCBI Gene Expression Omnibus (accession no: GSE15848) |
|---|---|---|---|---|
| Rai M, Soragni E, Jenssen K, Burnett R, Herman D, Coppola G, Geschwind DH, Gottesfeld JM, Pandolfo M | 2008 | HDAC Inhibitors Correct Frataxin Deficiency in a Friedreich Ataxia Mouse Model | https://www.ncbi.nlm.nih.gov/geo/query/acc.cgi?acc=GSE10745 | Publicly available at the NCBI Gene Expression Omnibus (accession no: GSE10 745) |
| Coppola G, Burnett R, Perlman S, Versano R, Gao F, Plasterer H, Rai M, Saccá F, Filla A, Lynch DR, Rusche JR, Gottesfeld JM, Pandolfo M, Geschwind DH | 2011 | A Gene Expression Phenotype In Lymphocytes From Friedreich's Ataxia Patients | https://www.ncbi.nlm.nih.gov/geo/query/acc.cgi?acc=GSE30933 | Publicly available at the NCBI Gene Expression Omnibus (accession no: GSE30 933) |

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
