## [Decision Letter]

Thank you for submitting your article "Inducible and reversible phenotypes in a novel mouse model of Friedreich's Ataxia" for consideration by *eLife*. Your article has been favorably evaluated by a Senior Editor and three reviewers, one of whom, J. Paul Taylor (Reviewer #1), is a member of our Board of Reviewing Editors.

The reviewers have discussed the reviews with one another and the Reviewing Editor has drafted this decision to help you prepare a revised submission.

Summary:

An important, unanswered question that has important implications for treating neurodegenerative diseases is the extent to which neurological deficits reflect reversible CNS dysfunction vs. irreversible neuronal loss. This question is relevant to all neurodegenerative diseases, though the answer may differ amongst them. The most commonly inherited ataxia, Friedreich's ataxia, results from partial loss of function mutations in FXN – most often from pathological expansion of an intronic GAA repeat. Knockout of FXN in mice results in embryonic lethality, but disease features have been recapitulated in mice through various strategies to partially and permanently reduce FXN expression.

Here the investigators established dose-dependent, conditional knockdown of FXN expression in a wide variety of tissues. This approach recapitulates many of the major features of Friedreich's ataxia, including behavioral motor and cardiac defects, as well as histopathological features in cardiac muscle, brain and peripheral nervous system. Importantly, the authors illustrate that these features are reversible upon restoration of FXN expression. Moreover, the authors provide detailed characterization of gene expression from cerebellum, heart and DRG to define a molecular signature associated with these behavioral and histopathological features, and this signature is also reversed upon restoration of FXN expression.

The reviewers were in agreement that this is a thoroughly characterized, robust model of Friedreich's ataxia that is used to illustrate a surprising extent of reversibility of disease-relevant features. However, the reviewers recommend that several issues be addressed prior to publication.

Essential revisions:

1) A controversial point in the FRDA field is related to the elevation of ROS in *Frataxin* loss animal models. The authors should measure ROS levels in the FRDA kd mice at two different time points and document if ROS is elevated or not.

2) In their TEM images, the authors observed lipid droplet-like empty vesicles. Navarro et al., (2010; HMG) and Chen et al. (2016b; *eLife*) have documented lipid droplet accumulations in fly nervous systems. The authors should refer to these papers as it strengthens their argument that they are indeed LD. The authors should also do a Nile Red stain to further strengthen their observations and show that they are probably LD.

3) Previous work in flies and mice and human has shown that PDK1 is hyperphosphorylated and that *Mef2* activity is upregulated in mice as well as hearts of patients (Chen et al., 2016a and b; *eLife*). None of their data related to transcription suggest that these genes are altered at the transcriptional level. They should discuss this and it would seem opportune to do a PDK1 Western blot to see if the same pathway is affected in their mice as what was observed in patients and CRISPR FXN mice.

4) Given the unavoidable inherent off-target possibility associated with shRNA knockdown approaches, the authors should comment on this possibility, or, ideally, check or test whether any transcripts with similar target sequences were not affected.

---

## [Author Response]

Essential revisions:1) A controversial point in the FRDA field is related to the elevation of ROS in Frataxin loss animal models. The authors should measure ROS levels in the FRDA kd mice at two different time points and document if ROS is elevated or not.

In this revised version, we have included new data showing that the reactive oxygen species (ROS) levels are not elevated in adult mice after Fxn knockdown (Figure 6—figure supplement 3). For quantification of ROS, we measured the levels of two markers of oxidative stress, 3-nitrotyrosine (3NT) and 4-hydroxy-2-nonenal (4-HNE), in brain, liver and muscle samples from the control and Fxn knockdown mice at 12 and 20 weeks. We did not observe an increase in these markers levels after Fxn knockdown at either stage. Next, we utilized the Oxyblot protein oxidation detection kit to detect the carbonyl groups introduced into proteins by increases in oxidative stress. Oxyblot analysis of protein oxidation in the brain and liver of Fxn knockdown and control mice also displayed no significant changes in the levels of 2,4-dinitrophenylhydrazone (DNP-hydrazone). These findings suggest that in adult mice, Fxn knockdown does not elevate ROS, which is consistent with the findings of Chen et al. 2016.

2) In their TEM images, the authors observed lipid droplet-like empty vesicles. Navarro et al., (2010; HMG) and Chen et al. (2016b; eLife) have documented lipid droplet accumulations in fly nervous systems. The authors should refer to these papers as it strengthens their argument that they are indeed LD. The authors should also do a Nile Red stain to further strengthen their observations and show that they are probably LD.

We thank the reviewer for this comment, and now cite Chen et al. in this regard. We also performed Nile Red staining in liver, heart, DRGs and heart samples from control and Fxn knockdown animals at week 20 (Figure 5—figure supplement 1). We observed strong staining in the liver samples in both transgenic and control animals indicative of lipid deposition. However, we did not observe positive staining in the heart, DRGs and spinal cord samples in either control and transgenic animals. Quantification of the number of lipid droplets in the liver samples did not show significant differences when comparing the control and transgenic animals. Additional experiments will be needed to identify the origin/composition of the electron-light or “empty” vesicles which we observed in the EM images of the transgenic animal DRGs.

3) Previous work in flies and mice and human has shown that PDK1 is hyperphosphorylated and that Mef2 activity is upregulated in mice as well as hearts of patients (Chen et al., 2016a and b; eLife). None of their data related to transcription suggest that these genes are altered at the transcriptional level. They should discuss this and it would seem opportune to do a PDK1 Western blot to see if the same pathway is affected in their mice as what was observed in patients and CRISPR FXN mice.

Thank you for bringing our attention to this recent finding. We performed focused re-analysis of our gene expression data which also supports that these genes are not altered at the transcriptome level. So, as suggested, we performed Western blot analyses to evaluate the phosphorylated levels of PDK and performed RT-PCR analyses to measure the top five candidate target genes of *Mef2* found in Chen et al. (2016b; *eLife*) (Figure 6—figure supplement 2). We utilized the same antibody (Cell signaling, RRID: AB_2161134) and primers utilized by Chen et al. (2016b; *eLife*) and have included these data in the supplement of the revised version of the manuscript. We observe that Fxn knockdown in adult mice does not alter the phosphorylation of S241 in the PDK1 activation loop (required for its activity) in brain, muscle, heart and liver samples obtained from Fxn knockdown mice.

Next, we analyzed the mRNA levels of top five candidate target genes of *Mef2* (Sgca, Hrc, Nr4a1, Myom1 and Tcap) in cerebellum and heart after Fxn knockdown at 20 weeks. All five genes have been reported to be overexpressed in Chen et al. (2016b; *eLife*). In two independent experiments utilizing 4 biological replicates, we found that Sgca (1 out of 5 genes tested) was significantly over-expressed after Fxn knockdown in the cerebellum. In the heart, we found Nr4a1 was significantly over-expressed and Hrc and Tcap were significantly down-regulated after Fxn knockdown (Figure 6—figure supplement 2). These results are consistent with tissue specific differential expression of these target genes due to Fxn knockdown, consistent with our global gene expression analyses as discussed in the manuscript. However, it does not directly replicate the findings of Chen et al. which could be because of the timing of knockdown in our model (not from birth). Further studies will be needed to see if embryonic or early postnatal chronic knockdown has a more definitive effect on this pathway. We discuss these results in the manuscript.

4) Given the unavoidable inherent off-target possibility associated with shRNA knockdown approaches, the authors should comment on this possibility, or, ideally, check or test whether any transcripts with similar target sequences were not affected.

Off-target effects are of concern in any experiment based on shRNA. To limit any off-target effect of our selected shRNA, initially we carefully selected our shRNA sequences targeting the Fxn gene through bio-informatic examination of sequence matches in the whole genome using BLAST. To increase confidence in our selected shRNA sequence, we performed independent experiments utilizing six different shRNA sequences targeting the Fxn gene (discussed in Materials and methods section and in Figure 1—figure supplement 1). We measured Fxn mRNA and protein levels in vitro after over-expression of these shRNA sequences. All our top six shRNA sequence displayed significant Fxn knockdown, showing that multiple shRNAs targeting the same gene (Fxn) produce comparable gene silencing efficacy.

To examine the off-target effect directly, we determined the shRNA specificity by looking at global changes in gene expression pattern during the induction of shRNA expression targeting Fxn in vitro and in our in vivo model. First, we screened and identified potential putative targets of the shRNA sequence (GGATGGCGTGCTCACCATTAA) in the whole mouse genome transcripts utilizing the BLASTN program. This identified 20 unique genes as potential targets in the mouse genome, which included Fxn gene as top hit with all 21 base pair matched as expected (Figure 1—figure supplement 2). The remaining potential targets had 16 to 13 base pair matches, which would not be predicted to lead to silencing. To address this directly, we examined the gene expression levels for all these potential target genes present in our microarray data after induction of the shRNA. Fxn was the earliest and consistent gene showing significant down-regulation and we did not observe any changes in the RNA levels of other potential targets at any stage in our gene expression analysis (Figure 1—figure supplement 2), indicating the absence of significant off-target effects. A brief discussion of these points has been incorporated in the Materials and methods section of the current version of the manuscript, and we have added a few sentences to the Discussion.